# Structural basis of VCP-VCPIP1-p47 ternary complex in Golgi maintenance

Binita Shah [1,2], Moritz Hunkeler [1,2], Ariana Bratt[1,2], Hong Yue [1,2], Isabella Jaen Maisonet [1,2], Eric S. Fischer [1,2] ✉ & Sara J. Buhrlage [1,2] ✉

VCP/p97 regulates a wide range of cellular processes, including post-mitotic Golgi reassembly. In this context, VCP is assisted by p47, an adapter protein, and VCPIP1, a deubiquitylase (DUB). However, how they organize into a functional ternary complex to promote Golgi assembly remains unknown. Here, we use cryo-EM to characterize both VCP-VCPIP1 and VCP-VCPIP1-p47 complexes. We show that VCPIP1 engages VCP through two interfaces: one involving the N-domain of VCP and the UBX domain of VCPIP1, and the other involving the VCP D2 domains and a region of VCPIP1 we refer to as VCPID. The p47 UBX domain competitively binds to the VCP N-domain, while not affecting VCPID binding. We show that VCPID is critical for VCP-mediated enhancement of DUB activity and proper Golgi assembly. The ternary structure along with biochemical and cellular data provides new insights into the complex interplay of VCP with its co-factors.

The AAA⁺ ATPase valosin-containing protein (VCP)/p97, known as cdc48 in *Saccharomyces cerevisiae*, is an essential and highly conserved protein involved in numerous cellular functions, such as regulating protein homeostasis[1], endolysosomal trafficking[2], membrane remodeling[3,4], chromatin regulation[5,6], autophagy[7–9], Golgi formation[10], and others[11–13]. Its core function is to unfold ubiquitylated proteins in an ATP dependent fashion[14–17]. VCP achieves its versatility to facilitate such diverse cellular functions by engaging many different cofactors[18], including adapter proteins that bind to VCP and mediate its interactions with substrates and/or additional binding partners[19–24]. How adapter binding governs VCP function has been extensively studied in the context of its role in the ubiquitin proteasome system (UPS)[25–27]. Structural and biochemical studies have played a significant role in providing a detailed understanding of how VCP unfolds ubiquitylated substrates by capturing various states of this AAA⁺ ATPase, with and without binding partners and substrates[15,17,25,28], motivating development of VCP ATPase inhibitors, which have advanced to clinical investigation. Less is known about how VCP in conjunction with multiple binding partners engages in a wide range of activities outside the UPS[27]. VCP binding partners can be categorized into three types: substrate recruiting cofactors, substrate processing cofactors and regulatory cofactors[29]. Traditionally, these cofactors have a VCP

interacting motif such as UBX or PUB domain that interacts with substrates, in most cases[18,20,23,30–32]. Studies using proteomics have uncovered new roles for VCP in ciliary biogenesis, endosomal sorting and connections to cullin-ring ligases while also providing insight into the highly dynamic nature of the interaction between VCP and its binding partners[31]. Understanding the role of VCP mechanistically and structurally with various binding partners will help to shed light on VCP regulation and may allow for better targeted therapy development[33]. Specifically, how VCP cooperates with the deubiquitylating enzyme (DUB) valosin contain protein p97/p47 interacting protein 1 (VCPIP1) and, the adapter, p47 in facilitating Golgi reassembly[34,35], two different binding partners each with UBX domains, remains poorly understood at the molecular level[36,37].

Previous studies have established that post-mitotic Golgi reassembly requires presence of VCP, as well as two adapter proteins, VCPIP1[38], and p47[34,35]. VCPIP1, also known as VCIP135, is part of the ovarian tumor protease (OTU) family of DUBs, an essential family of enzymes that remove ubiquitin tags from proteins to regulate protein homeostasis and function[39,40]. In Golgi, mono-ubiquitylation of syntaxin-5 (Syn5), a t-SNARE protein essential for Golgi reassembly[41], allows for the regulation of both assembly and disassembly processes during cell division. VCPIP1, in complex with VCP and p47,

¹Department of Cancer Biology, Dana-Farber Cancer Institute, Boston, MA, USA. ²Department of Biological Chemistry and Molecular Pharmacology, Harvard Medical School, Boston, MA, USA. ✉e-mail: Eric_Fischer@dfci.harvard.edu; SaraJ_Buhrlage@dfci.harvard.edu

deubiquitylates Syn5 thereby controlling the affinity of the SNARE proteins[42], allowing deubiquitylated Syn5 to bind to v-SNARE Bet1, resulting in membrane fusion and Golgi reassembly[34,43,44].

To understand how VCP engages with VCPIP1 and p47, we determined high resolution cryogenic electron microscopy (cryo-EM) structures of both VCP-VCPIP1 and VCP-VCPIP1-p47 complexes capturing interaction states at residue resolution despite the high conformational flexibility. Our structures reveal that VCPIP1 exhibits a bivalent binding mode through two distinct regions of VCP. The VCPIP1 ubiquitin regulatory X (UBX) domain engages the N-domain of VCP, which is a well-established primary site of adapter protein binding[34]. Interestingly, we also identified a region on VCPIP1 that engages VCP at its C-terminal domain (D2 domain)[45], which we named "VCP interacting domain" (VCPID) after validating its functional relevance. We found that while the UBX domain was essential for binding, VCPID was important to promote DUB activity. To put the VCP-VCPIP1 structure in context with the critical VCP adapter p47, we determined the structure of a VCP-VCPIP1-p47 ternary complex, which reveals that the p47 UBX domain outcompetes the VCPIP1 UBX domain on the VCP N-domain, which may suggest dynamic competition for N-domain binding, as seen with other VCP cofactors[18,24,46] and in other VCP processes. Lastly, we show in cell-based studies that VCPIP1 UBX domain, VCPID and its DUB activity are all required for previously reported functions of VCPIP1[34,38] in Golgi maintenance. In this work, we demonstrate how VCPIP1 bivalently binds to VCP via the UBX domain and the newly characterized VCPID and how these domains along with its DUB activity are crucial for Golgi assembly.

## Results

### VCP interacts with VCPIP1 through multivalent interactions

To reconstitute the VCP-VCPIP1 complex for structural studies, we co-expressed and purified full-length VCPIP1 and VCP (Supplementary Fig. 1a, see Fig. 1a for schematics of domain organization indicating domain boundaries) from Expi293 cells with no addition of ATP or ATP analogs (see "Methods"). Two-dimensional (2D) class averages from initial cryo-EM data sets revealed the typical hexameric structure of VCP with additional blurred density visible at the carboxy (C)-terminal domain of VCP, indicating significant conformational heterogeneity of this region (Supplementary Fig. 1b). To stabilize the complex, we used BS3 chemical crosslinking prior to grid preparation and cryo-EM data collection (Supplementary Fig. 1c). From this sample, initial 3D reconstructions revealed two binding sites of VCPIP1 on VCP (Fig. 1b). We obtained a consensus reconstruction of the VCP-VCPIP1 complex at an overall resolution of 2.3 Å (see "Methods" and Supplementary Figs. 2, 3 for details). To better resolve these regions, we used symmetry expansion followed by focused refinement, with masks focusing on the N- and C- terminal ends of VCP. This enabled us to obtain improved maps of specific subsections, including a 2.9 Å map resolving three VCPIDs bound to the D2 domains of VCP without symmetry expansion, a 2.9 Å map of one VCPID with stalk region bound to the bottom interface of VCP D2 dimer and a 3.1 Å map of the VCPIP1 UBX domain bound to the N-terminal domain of VCP, both with symmetry expansion (Supplementary Fig. 3a–d, see Supplementary Table 1 for details).

Of the two identified binding regions, we were able to clearly assign the VCPIP1 UBX domain (aa 773–852) and identified the larger interacting domain to the region spanning the stalk region (aa 356–588) and the domain of unknown function as VCPID (aa 589 – 666), which we newly characterize in this study (Supplementary Fig. 1d). The VCPIP1 UBX domain is bound to the flexible N-domain of VCP limiting the resolution of the initial reconstructions which was addressed with C6 symmetry expansion to obtain a high-resolution map and build an atomic model (Fig. 1c). We observed that the VCPIP1 UBX domain folds into an archetypal UBX domain structure. The interaction interface is mainly formed by F821 on the UBX domain, buried in a hydrophobic pocket formed by VCP V38, I70, and L72 (Supplementary Fig. 1e). Additional contacts are made between VCPIP1 P823 and VCP A142. We also identified a possible cation-π interaction between VCPIP1 UBX domain R778 and VCP F52 (Fig. 1c).

In addition to the anticipated VCPIP1 UBX domain interaction with the VCP N-domain, we observed density corresponding to three VCPIP1 interfacing with the D2 domain dimers of VCP (Fig. 1b), which we later assigned to the VCPID domain of VCPIP1. VCPID consists of a β-hairpin loop (aa 590–609) and two alpha helices (α1 (aa 622–633) and α2 (aa 640–654)) (Fig. 1d, and Supplementary Fig. 1d) and was determined to have no domain conservation amongst the PDB[47,48]. The observed stoichiometry between VCP and the VCPID domain of VCPIP1 is 2:1 with each VCPID contacting two D2 domains (D2 chain 1 and D2 chain 2, Fig. 1b). The VCPIP1 VCPID interacts with the dimer through its alpha helices, while the β-hairpin loop connects to the VCPIP1 stalk region (Fig. 1d). Notably, VCPID Y623 makes π-π interactions with F758 from VCP D2 chain 1 and engages through insertion in a hydrophobic pocket formed by D627, K754 and T761 on the same chain (Supplementary Fig. 1f). VCPID N621, D627 and K754 also interact with VCP M757, L652 and Q645, respectively (Fig. 1d). The primary contacts of VCPID with the VCP D2 chain 2 are formed through insertion of VCPID F638 into a hydrophobic pocket formed by P626, Y755, and F758 (Fig. 1d, Supplementary Fig. 1f). This interface is further stabilized by a salt bridge formed between VCP K754 (D2 Chain 2) and VCPID E641 (Fig. 1d). The three VCPIDs interact exclusively with VCP and make no inter-domain contacts (Supplementary Fig. 2).

We observed variable quality of density for the VCPIDs, more specifically the β-hairpin loops, suggesting that there is compositional and/or conformational heterogeneity in the binding region. To enrich the dominant conformation, we again used focused classifications and refinements and were able to visualize additional density corresponding to the stalk region that binds to the VCPIP1 OTU domain (2.9 Å resolution; Fig. 1b). Further, this allowed us to visualize the movement of the OTU domain (aa 215–355) using three-dimensional variability analysis (see "methods"), which contains the catalytic cysteine (C219) responsible for DUB activity, and the stalk region that connects to VCPID (Supplemental Movie 1). The flexible nature of the stalk and the OTU domain prevented us from obtaining a high-resolution structure for these regions of VCPIP1, thus a portion of the stalk and all of the OTU domain are excluded from model building (Supplementary Fig. 2). Nonetheless, we analyzed different 3D classes to evaluate the trajectory of the stalk and the OTU domain. From the density visible, the stalk region moves ~4 Å and the OTU domain moves ~13.5 Å from the first to the last state (Fig. 1e).

Taken together, our cryo-EM structure of shows VCPIP1 bound to VCP in a 1:2 stoichiometry, although we cannot fully exclude the possibility that additional VCPIP1 could engage through the N-domain only. Each VCPIP1 molecule makes two distinct contacts with VCP, one mediated by the UBX domain that engages the N-domain of a VCP protomer, and one with the VCPID that forms contacts with D2 domains of two adjacent VCP protomers. These interactions position the catalytic domain of VCPIP1 in close proximity to the end of the central pore of VCP. We also observed that the complex is highly dynamic, including within the N-domain of VCP and throughout VCPIP1.

### The UBX domain anchors VCPIP1 to VCP

Our structural analysis showed that both the UBX domain and VCPID domains of VCPIP1 bind VCP. To dissect their roles further, we measured binding affinities of wild type VCPIP1 (VCPIP1$^{WT}$) and different VCPIP1 domain deletion constructs to VCP: VCPIP1$^{\Delta VCPID}$ (aa Δ589 – 666); VCPIP1$^{\Delta UBX}$ (aa Δ773 – 852); and VCPIP1$^{\Delta VCPID\ \Delta UBX}$ (aa Δ589–666, Δ773–852) (Fig. 2). Using a time-resolved fluorescence resonance energy transfer (TR-FRET) assay[49], we observed that the two constructs with an intact UBX domain, VCPIP1$^{WT}$ and VCPIP1$^{\Delta VCPID}$, bind to VCP with

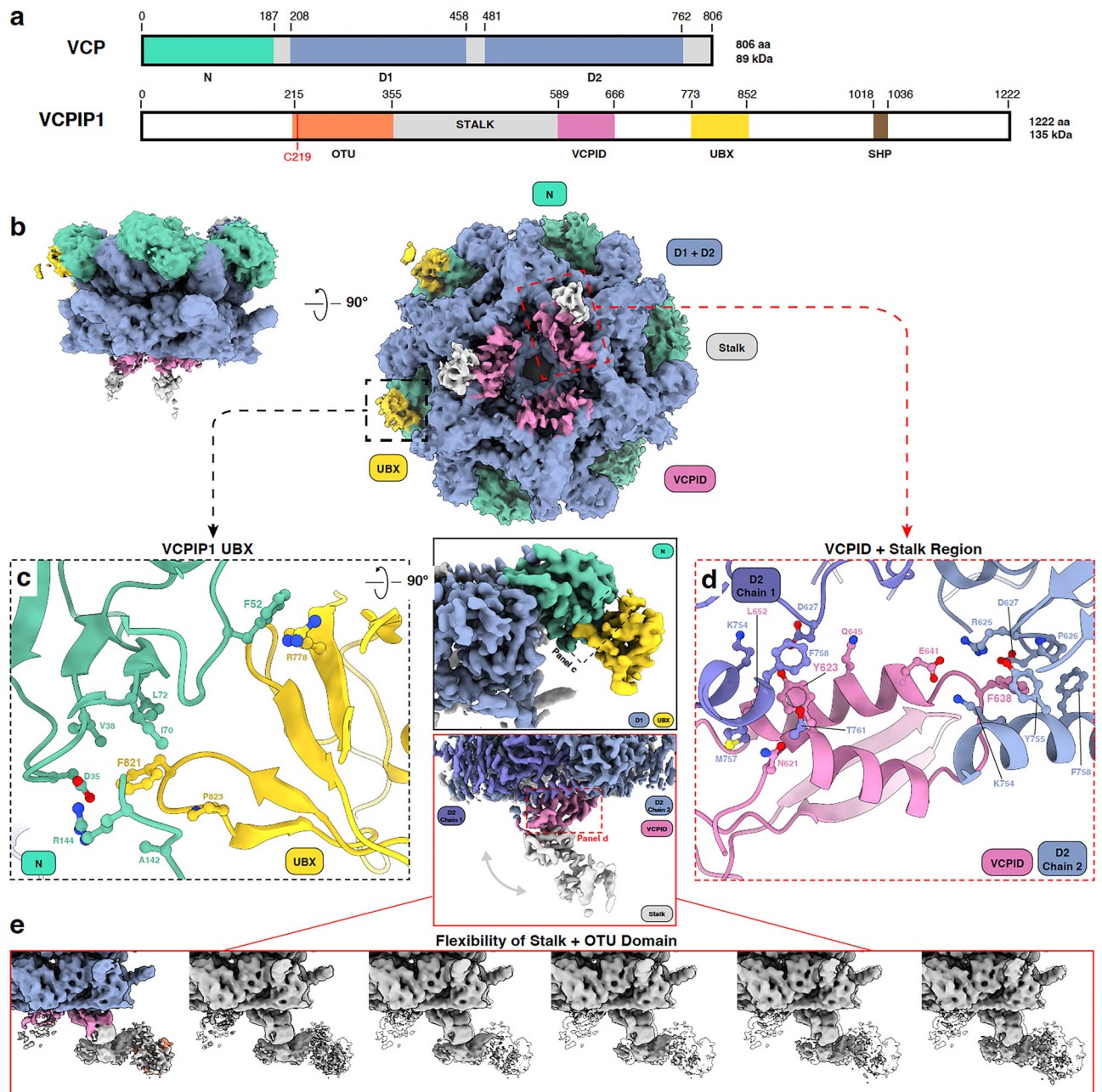

**Fig. 1 | Cryo-EM structure depicting VCPIP1 as a multivalent binder of VCP.**
**a** Domain organization of VCP (top) indicating the three main domains: N-domain (green) and two ATPase domains (D1 and D2, blue), and VCPIP1 (bottom), indicating the OTU deubiquitylase (DUB) domain (orange) with the location of the catalytic cysteine (C219) (red), stalk region (grey), newly annotated VCPID (pink) and the UBX domain (yellow). **b** Overall structure of the VCP-VCPIP1 complex in two orientations, color coded using the same colors as in (**a**), with two zoomed in regions focusing on two areas of VCP where additional density assigned to different regions of VCPIP1 are present. (**c**) Zoom in on the VCP N-domain interface with the VCPIP1 UBX domain displaying key interactions and features. (**d**) Zoom in on the VCP D2-domain dimer and its interaction with the newly defined VCPID. (**e**) Heterogeneity in our cryo-EM maps caused by the dynamic nature of VCPIP1 stalk and OTU domain (state 1 colored and then depicted in an outline, subsequent states in grey to highlight movement).

comparable apparent affinity ($K_{D,app}$ = 79.5 nM and $K_{D,app}$ = 67.9 nM, respectively) (Fig. 2a, b). In contrast, the constructs lacking the UBX domain, VCPIP1$^{\Delta UBX}$ and VCPIP1$^{\Delta VCPID\ \Delta UBX}$, did not show quantifiable affinity for VCP (Fig. 2c, d), while some possibly non-specific binding presumably from interactions with VCPIP1 VCPID and/or SHP box was observed. These results reveal that the UBX domain of VCPIP1 is critical for VCP binding. This is in agreement with previous studies that identified the C-terminal region of VCPIP1, where the UBX domain is located, as critical for VCP binding[34]. Based on these studies, we

propose that VCPIP1 UBX domain binding serves to anchor VCPIP1 to VCP, allowing VCPID to reach and engage the D2 domain region of VCP.

Since previous studies have shown that bound nucleotide influences the conformation of the N-domain[16,29,50], we asked whether it may also affect binding of VCPIP1 to VCP. We utilized the TR-FRET binding assay to assess the binding of VCPIP1to VCP using a TR-FRET assay in the presence of the nucleotides / nucleotide analogs ATP, ADP, AMP and ATPγS (see Supplementary Fig. 8). We find that indeed the

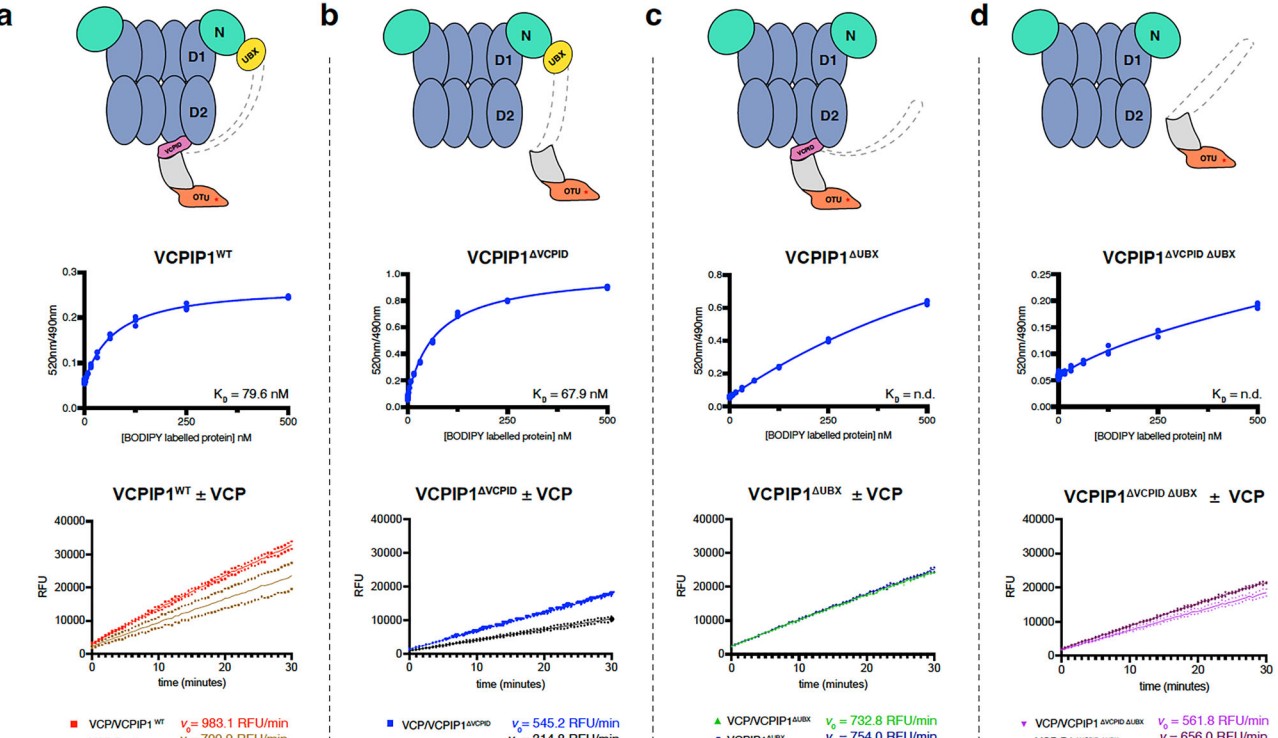

**Fig. 2 | Characterization of VCPIP1 domain functionality in complex with VCP.** First row shows cartoon depictions of VCP in complex with VCPIP1 containing various domain truncations, second row depicts TR-FRET binding results and last row depicts DUB activity results at 62.5 nM enzyme concentration. Dashed grey lines are used to organize data with corresponding VCPIP1 truncation, **a** VCPIP1[WT], **b** VCPIP1[ΔVCPID], **c** VCPIP1[ΔUBX] and (**d**) VCPIP1[ΔVCPID ΔUBX], with the corresponding biochemical data.

bound nucleotide influences the affinity of VCPIP1 to VCP with either buffer control or AMP exhibiting the tightest binding with $K_{D,app}$ of = 16.5 nM or 16.1 nM, respectively. In contrast, addition of ATP, ADP or the non-hydrolysable ATP analog ATPγS significantly weakens the interaction ($K_{D,app}$ = 68.3 nM for ATP, 67.6 nM for ADP and 63.9 nM for ATPγS; we note that there are differences between the apparent affinities of these experiments to the previously determined affinities attributed to different preparations of the proteins leading to batch effects but not impacting within experiment comparisons). It has been previously observed that the N-domains of VCP in the presence of AMP, similar to the absence of exogenous nucleotide, adopt a heterogeneous conformation with the domains in both the up and down positions[16,50], while the N-domains have been found to be in the up-position in the presence of ATP and ATPγS[16,51]. Lastly, ADP has been found to result in the N-domains in the down-position[50]. These data suggest that for VCPIP1 binding to VCP, locking the N-domains in either the up- or down- positions presents an energetic barrier which may be due to the conformation of VCPIP1 bound VCP being slightly different to the locked states. Taken together, these data not only reiterates that the UBX domain of VCPIP1 is the main binding domain to VCP but also provides further evidence that VCPIP1 UBX binds to the N-domain of VCP in both the up and down conformation, confirming what we see using cryo-EM (see intermediate densities in Supplementary Figs. 2, 6). These findings are consistent with a recent report by *Vostal* et al.[52] that VCPIP1 UBX domain can sample multiple conformations of the N-domains and hence a flexible state of the N-domain would be preferential.

## VCPID supports DUB activity of VCPIP1
We next asked whether complex formation with VCP influences the deubiquitylation activity of VCPIP1 using ubiquitin-rhodamine 110 (Ub-Rho) as a generic DUB substrate to avoid substrate specific effects. We

measured the initial velocities of the VCPIP1-mediated deubiquitylation reaction with and without full length VCP. In agreement with the literature[39,53], we see that VCPIP1[WT] is an active DUB ($v_O$ = 700.9 RFU/min; Fig. 2a, and Supplementary Fig. 4a). Removal of the UBX domain has no major effect on the initial velocity of the deubiquitylation reaction ($v_O$ (VCPIP1[ΔUBX]) = 754.0 RFU/min; Fig. 2c). However, removing VCPID decreased the rate of ubiquitin hydrolysis significantly ($v_O$ (VCPIP1[ΔVCPID]) = 314.8 RFU/min) (Fig. 2b). Surprisingly, the loss in activity was partially restored in the double deletion of UBX domain and VCPID ($v_O$ (VCPIP1[ΔVCPIDΔUBX]) = 656.0 RFU/min; Fig. 2d), suggesting a potential auto-inhibitory mechanism involving the UBX. The presence of VCP increased the rate of deubiquitylation displayed by VCPIP1[WT] ($v_O$ = 983.1 RFU/min; Fig. 2a) and VCPIP1[ΔVCPID] ($v_O$ = 545.2 RFU/min; Fig. 2b) but had no effect on the activity of VCPIP1[ΔUBX] ($v_O$ = 732.8 RFU/min; Fig. 2c) and VCPIP1[ΔVCPID ΔUBX] ($v_O$ = 561.8 RFU/min; Fig. 2d).

Collectively, these data suggest that binding to VCP enhances the rate of the deubiquitylation reaction catalyzed by VCPIP1[WT] (Supplementary Fig. 4b). We observed that VCPID removal has the largest effect on the rate of the DUB activity, suggesting that this domain either helps in orienting the active site towards the substrate or allosterically activates DUB activity, both in the presence and absence of VCP (Supplementary Fig. 4c). Importantly, removal of the UBX domain resulted in constructs that are not sensitive to the presence of VCP, which is in line with reduced affinity for VCP as determined using a TR-FRET assay (Fig. 2a–d). Thus, for the highest level of VCP-dependent enhancement of deubiquitylation activity, both the UBX domain and the VCPID of VCPIP1 need to be present.

## Structure of the VCP-VCPIP1-p47 ternary complex
The activity of VCP in Golgi reassembly has been shown to require involvement of both VCPIP1 and p47[34] (Fig. 3a). To provide a structural basis of how these two cofactors simultaneously interact with VCP and

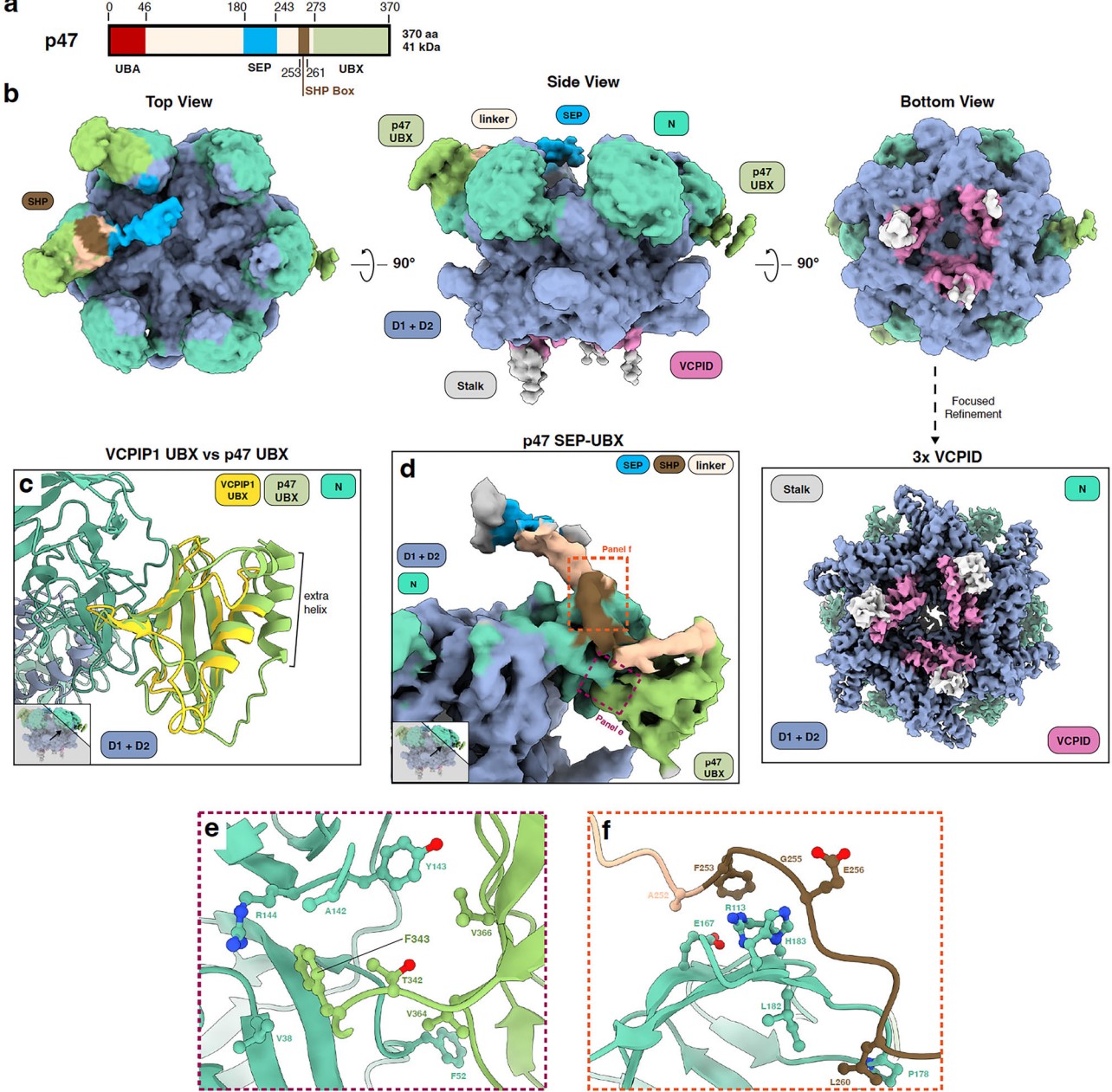

**Fig. 3 | Cryo-EM structure of VCP bound to VCPID and adapter protein p47.**
**a** Domain organization of p47 showing the presence of its major domains: ubiquitin associated (UBA) (red), SEP domain (blue), SHP box (brown), UBX domain (lime green) and the connecting linker (beige). **b** Low pass filtered (6.0 Å) map of VCP-VCPIP1-p47 ternary complex depicted in top, side and bottom views with a 90° rotation. **c** Comparison of VCPIP1 UBX and p47 UBX domains. **d** Zoom in on density for p47 bound to VCP N-domain. **e** zoom in of interactions formed by p47 UBX domain and VCP N-domain. **f** zoom in on interactions formed between p47 SHP binding site and linker region and VCP N-domain.

a framework for understanding this process, we determined the first structure of VCP in a ternary complex with VCPIP1 and p47 using cryo-EM (Supplementary Fig 5a–c). After initial rounds of 3D classification, we were again able to observe density in the two binding regions on VCP, similar to what we observed for the VCP-VCPIP1 complex (see "Methods" and Supplementary Figs. 6, 7 for details). We used a combination of symmetry expansion and focused refinement to zoom in on these regions. We obtained a 3.1 Å reconstruction revealing three VCPIDs bound at the C-terminal end of VCP without symmetry expansion, and a 4.0 Å reconstruction of the N-domain region with symmetry expansion. The binding mode of the three VCPIDs bound to the D2 dimer is indistinguishable from the binding mode observed in

the VCP-VCPIP1 complex, with the stalk regions connected to each VCPID clearly visible (Fig. 3b). Since the interactions between VCP and the VCPID in all structures are unchanged, we conclude that p47 binding does not affect how VCPIP1 engages the D2 domains of the VCP hexamer.

We next focused on the UBX domain observed on VCP N-domains. We find that there is clear density for three UBX domains, with two bound to N-domains in the up-position and one bound with N-domain in the down-conformation (Supplementary Fig. 5d, see "Methods", Fig. 3b). Although the UBX domains from VCPIP1 and p47 share structural homology (Fig. 3c), the p47 UBX domain has an additional helix (aa 273–286), which allowed us to assign all three visible UBX

domains as originating from p47 (Supplementary Fig. 5e–f). Using AlphaFold3[54] predictions of VCP N and D1 domain bound to p47, we were able to fit in the highest ranked predicted model into our density (Supplementary Fig. 5g, see "Methods"). Compared to the VCPIP1 UBX domain, the p47 UBX domain binds to a partially overlapping area of the VCP N-domain (Fig. 3d). It also uses a phenylalanine, F343, to occupy a hydrophobic pocket within the N-domain defined by VCP's V38, A142, Y143, and R144 (Fig. 3e, and Supplementary Fig. 5h). Additional interactions are mediated by V364 and V366 of p47 UBX domain that engage with F42 and Y143 of the VCP N-domain, respectively (Fig. 3e). The extra helix of the p47 UBX domain is connected to a short loop (aa 244–272) which is connected to the SEP domain (aa 180–243) through a long linker that includes the SHP box (aa 253–261) (Fig. 3d), a motif previously reported as critical for VCP binding[24,45,55]. We observe significant, yet lower resolution, density for the linker to the SEP domain with the SEP domain potentially hovering over the central pore of VCP (Fig. 3b, see "Methods"). The SHP box acts a secondary binding site for p47 UBX domain as it has residues that interact with the top of the VCP N-domain, unlike VCPIP1 UBX domain (Fig. 3d, f). Specifically, L260 of p47 further stabilizes P178 and L182 on the VCP N-domain, G255 and E256 interact with H183 on VCP and F253 SHP approaches R113 and E167 (Fig. 3f). Lastly, there are additional contacts through A252 right next to the SHP motif, which interacts with E167 on VCP (Fig. 3f).

To measure and compare the affinities of VCPIP1 and p47 for VCP, we developed a TR-FRET displacement assay (Supplementary Fig. 5i). Unlabeled p47 and VCPIP1[WT] were used to compete out a pre-formed complex of terbium-chelate labeled VCP and BODIPY-VCPIP1[WT] (see "Methods"). We find that p47 ($IC_{50}$ = 20.4 nM) has a slightly lower $IC_{50}$ than VCPIP1[WT] ($IC_{50}$ = 74.54 nM). Compared to the single SHP box that VCPIP1 contains, p47 has a secondary SHP box on the N-domain of VCP, one observed in the ternary structure and one that is not, possibly explaining the difference in affinity between VCPIP1 and p47 for VCP. However, given that the apparent affinities are similar, that p47 and VCPIP1 UBX domains bind to overlapping locations on VCP[18], and that VCPIP1 UBX domain is the anchor point for VCP engagement based on our data (Fig. 2), it is likely that both p47 UBX domain and VCPIP1 UBX domain bind to VCP in a dynamic manner, which may be further modulated in the presence of substrate. Taken together, the cryo-EM structure of VCP-VCPIP1-p47 complex reveals how VCP is able to engage with p47 and VCPIP1 simultaneously and maps key points of interaction between VCP and the two adapter proteins involved in mediating Golgi reassembly. The binding appears to be dynamic and not mutually exclusive, suggesting multiple regulatory mechanisms.

## VCPIP1 interactions with VCP and catalytic activity are required for effect on Golgi

Next, we asked whether the interaction of VCPIP1 with VCP is necessary for the previously reported role of VCPIP1 in Golgi reassembly. We developed a cellular system to assess the role of different VCPIP1 domains in which the knock-out of VCPIP1 resulted in impaired appearance of Golgi that can be rescued by add-back of exogenous VCPIP1. We first used CRISPR/Cas9 to engineer A2058 cells with bi-allelic knock-out (KO) of VCPIP1. To observe Golgi, we stained cells with nuclear stain (DAPI) and an antibody specific to the protein that stabilizes the Golgi matrix (GM130)[56]. Compared to the more expansive Golgi matrix formed in the wild type A2058 cells, VCPIP1 KO cells exhibited a condensed matrix as determined by quantifying Golgi area, as observed by confocal microscopy (Fig. 4a VCPIP1 WT vs VCPIP1 KO and methods), which we took as a qualitative readout for the previously reported inability to reassemble the Golgi matrix in absence of VCPIP1[34]. Next, both A2058 VCPIP1 WT and KO cells were transiently transfected with an empty vector as controls, and KO cells were further transfected with either VCPIP1 wild type or mutant constructs (VCPIP1[ΔVCPID], VCPIP1[ΔUBX], VCPIP1[ΔVCPID ΔUBX], and VCPIP1[C219A]) (Fig. 4a). We

observe condensed Golgi staining in VCPIP1[KO], VCPIP1[EmptyVector], VCPIP1[ΔVCPID], VCPIP1[ΔUBX], VCPIP1[ΔVCPID ΔUBX] and VCPIP1[C219A] conditions indicated by a decrease in Golgi area (Fig. 4b). In contrast, transfecting the VCPIP1[WT] construct into VCPIP1 KO cells (VCPIP1 rescue) partially phenocopied features[38] of the WT VCPIP1 appearance indicated by expansive and dispersed Golgi matrix present next to the nucleus[57] (Fig. 4b) quantified as an increase in Golgi area. We note that the transient expression level of wild type VCPIP1 was significantly lower than endogenous VCPIP1 or most of the mutant constructs (Fig. 4a), likely attenuating the rescue effect. To also quantitatively assess the difference, the area of Golgi matrix was estimated by manually creating a freeform mask (Fig. 4c) encapsulating the Golgi signal as a measure for Golgi re-assembly (Fig. 4d, see "Methods"). In line with the qualitative observations, we find significant differences in area of Golgi per cell between WT and KO conditions, KO and VCPIP1 rescue condition and KO and VCPID domain truncation ($p < 0.0001$). There was no significant difference between KO and the UBX domain truncation, the double domain truncation or the catalytic cysteine mutant demonstrating that these truncations and mutations impair VCPIP1 function compared to KO phenotype. By visual inspection, VCPIP1[ΔVCPID] and VCPIP1[ΔUBX] constructs appear to have some rescue of WT phenotype but largely resemble the KO phenotype. The overall data together suggest that VCPID, UBX domain and the catalytic cysteine all significantly alter Golgi area and are essential for VCPIP1 activity in Golgi assembly.

## Discussion

VCP engages with a wide range of cofactor proteins to function in distinct cellular processes. However, our understanding of how VCP engages with different adapters in pathways outside the UPS remains poorly understood, for example the DUB VCPIP1. Here, we provide structural insights into how VCP simultaneously engages VCPIP1 and p47, two adapters that are critically involved in post-mitotic Golgi reassembly. Our structure of the VCP-VCPIP1 complex reveals that, unlike other structurally characterized VCP-adapters that show a single point of contact, VCPIP1 forms two distinct interfaces with VCP. This bivalent interaction is mediated by the well-characterized UBX domain bound to the N-terminus of VCP, and VCPID bound to the C-terminal VCP D2 domain. In the structure, we observed a 2:1 VCP/VCPIP1 stoichiometry, with each UBX domain engaging a single N-domain of VCP, and each VCPID engaging two D2 domains from neighboring VCP protomers. While the binding mode of VCPID enforces an upper limit to the simultaneously engaged VCPID to VCP as one VCPID per two protomers of VCP, the multivalent interaction encompassing UBX and VCPID, the competitive binding of different UBX containing co-factors, nucleotide state and possibly unknown factors support multiple possible stoichiometries under physiological conditions, which will require future studies to dissect. The bivalent mode of binding between VCP and VCPIP1 is further critical to understand how VCPIP1 can engage in its physiological complex with VCP and p47. During the preparation of our manuscript, two studies by Liao et al.[58] and Vostal et al.[52] also reported on VCP-VCPIP1 complexes describing overall similar arrangements of the complex and reporting several overlapping findings that mutually reinforce the work. Some differences may be explained by the use of different expression systems, which has been reported to result in differences in nucleotide state[59]. Using TR-FRET experiments (Supplementary Fig. 8), we assessed the effect of nucleotides on VCPIP1 binding and find that the absence of exogenous nucleotide behaves similarly to the addition of AMP in the complex, suggesting our complex is nucleotide free in line with not observing nucleotides bound in the structure. Although exogenous ATPs were not used for structural studies, VCP N-domains were seen in the up and down conformation for the VCP-VCPIP1 complex and further refinement with six-fold symmetry was done in both conformations, but there was substantially less UBX domain density bound in the up-

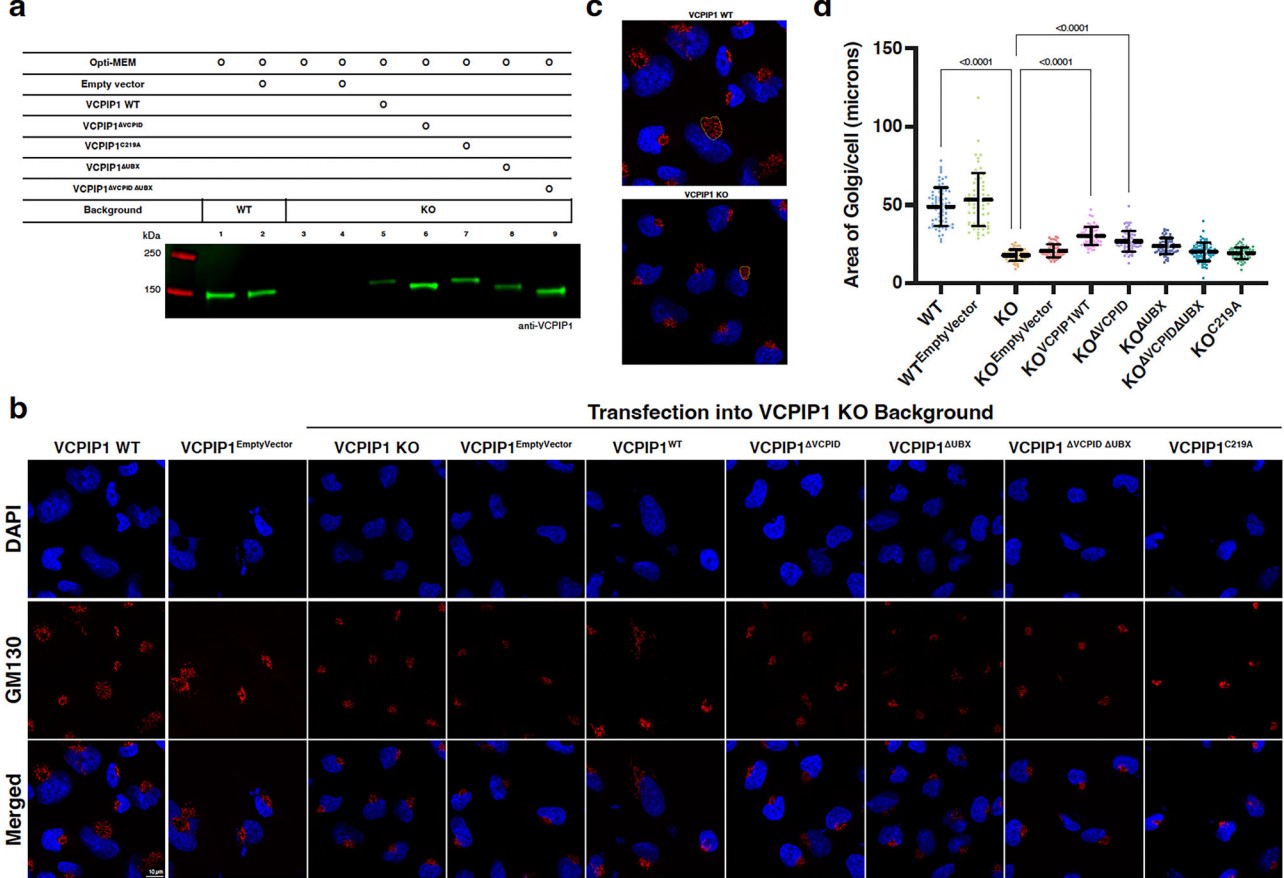

**Fig. 4 | VCPIP1 catalytic activity and interaction with VCP contribute to Golgi assembly. a** Representative western blot, blotting for VCPIP1, demonstrating successful transient transfection of VCPIP1 constructs in A2058 VCPIP1 KO cells. **b** Representative fluorescent images of A2058 WT and KO cells in which various VCPIP1 constructs were transiently transfected into the cells. Cells were stained with DAPI (first row) and GM130 antibody (second row). Both channels were merged as depicted in the third row. Scale bar is 10 μm (depicted in white). Images, biological triplicates each with technical duplicates, were taken on different days using the same imaging set-up, which may result in differences in apparent brightness. **c** Representative freeform mask (yellow) drawn in ImageJ Fiji around Golgi signal from one cell from a single merged image from WT and KO conditions are shown. **d** Quantification of area of Golgi per cell from various conditions are represented in a GraphPad Prism column chart. The data represented as mean ± SD from three biological replicates each with technical duplicates The *P* values were calculated using one-way ANOVA between KO and all other conditions using GraphPad Prism software ($p < 0.0001$).

position compared to the down-position (see intermediates in Supplementary Fig. 2).

An important insight to emerge from our structure, binding and activity studies is the involvement of different domains of VCPIP1 in its core activity. We provide evidence that while VCPIP1 UBX domain serves as an anchor for VCP binding, VCPID is required for VCP-mediated enhancement of VCPIP1 DUB activity. This model is further supported by sensitivity of VCPIP1 binding to VCP to nucleotide state (see Supplementary Fig. 8). In the context of Golgi assembly, phenotypically, we observed significantly smaller areas of Golgi within VCPIP1[ΔVCPID], VCPIP1[ΔUBX], VCPIP1[ΔVCPIDΔUBX] and VCPIP1[C219A] conditions suggesting that these regions and the DUB activity all have an important role in VCPIP1 function and its role in post-mitotic Golgi assembly. In prior reports, slight changes in Golgi architecture due to VCPIP1 mutations have been associated with unsuccessful reassembly. From images published of Golgi using transmission electron microscopy, cisternae appear structured in its assembled form but are broken down in its disassembled form[41]. Other groups have attempted to quantitatively assess the changes in Golgi by assigning centromeres, measuring distances and averaging[60], while others have used machine learning models by analyzing features such as orientation, perimeter and area[61]. The Golgi apparatus is a complex organelle with multiple components, such as cis and trans Golgi, and

studying it in greater detail would enable deep understanding of the intricate mechanisms. For the purpose of this study, we sought to use overall Golgi area as a proxy for VCPIP1 associated defects in Golgi assembly and we present a quantification methodology that accurately demonstrates the observed variance between treatment conditions. It is important to note that while this is likely due to defects in reassembly, this is not directly demonstrated but rather based on previous literature.

The structure of the VCP-VCPIP1-p47 complex presented here establishes how both VCPIP1 and p47 simultaneously bind to VCP to form a dynamic ternary complex crucial for proper Golgi assembly. This structure and accompanying biochemical data are of high significance as for example it helps to clarify the selective effects of CB-5083, an ATP-competitive VCP inhibitor previously tested in clinical trials (terminated for toxicity[62]) based on the hypothesis that VCP inhibition leads to an accumulation of misfolded proteins and proteotoxic stress-induced apoptosis in certain cancers. CB-5083 preferentially targets and has high specificity for the D2 domain of VCP[33,62,63] ($IC_{50} = 11$ nM)[62,64]. Interestingly, it has been shown that the binding of p47 to VCP decreases the potency of a D2 domain targeted inhibitor ~50 fold[65]. In contrast, an inhibitor that targets both the D1 and D2 domain has a ~4-6 fold decrease in potency with p47 bound[65] demonstrating inter domain communication[63], and possible

downstream effects on cellular functions including Golgi assembly[63]. Our work demonstrates a very dynamic nature of the VCP complexes, where different cofactors compete for binding and where their binding may favor one conformation over another, thus impacting potency of different inhibitors by allosteric mechanisms linking different domains. Additionally, our structures offer a blueprint for domain-specific targeting of VCP, including a possibility of designing protein-protein interaction inhibitors that would affect only Golgi assembly pathway, and not ubiquitin-dependent protein degradation and vice versa.

In addition to the significance of our results for future drug discovery efforts, our work also has clear implications for understanding VCP biology. Previous studies have demonstrated that VCP, in complex with VCPIP1, is necessary for cisternal regrowth[38], and that p47 may serve to recruit the VCP-VCPIP1 complex to the substrate[24,35,42]. In that model, mono-ubiquitylated Syn5 is recognized by the UBA domain of p47[35,42], which brings the substrate into the vicinity of VCPIP1 for deubiquitylation. Following deubiquitylation, Syn5 is free to bind Bet1 and form a SNARE complex resulting in membrane fusion, whereas VCPIP1 dissociates from VCP-p47, potentially driven by ATP hydrolysis induced conformational changes in VCP[42]. Therefore, based on this mechanistic proposal and our structural data, we argue that, in the context of post-mitotic Golgi assembly, VCP-VCPIP1-p47 complex behaves as a multicomponent DUB, rather than the classical VCP ATPase/unfoldase machinery. Additional studies will be needed to test this mechanistic model further.

In conclusion, we report six new models of VCP bound to VCPIP1 and in a ternary complex of VCP-VCPIP1-p47. The high resolution of our structures, along with a heterogeneity and dynamics aware processing compared to similar studies[52,66] enabled us to uniquely identify a high degree of dynamics and flexibility in VCP-VCPIP1 and VCP-VCPIP1-p47 complexes. We reveal a new, bivalent mode of engagement between VCP and VCPIP1, including the unique interface between the newly annotated VCPID of VCPIP1 and D2 domains from two VCP protomers. Given that these interactions are maintained in the context of the VCP-VCPIP1-p47 complex we report here, we expect that they play a key role in ensuring efficient Syn5 deubiquitylation for membrane fusion.

## Methods

### Cloning, protein expression and purification

Full-length VCPIP1 in pDEST_CMV with a N-terminal Strep tag was acquired from Wade Harper[67] and VCP was a gift from Nico Dantuma (VCP in pEGFP-N1 [Addgene plasmid #23971]). From there, the VCP coding sequence was moved into a modified pDARMO (pDarmo.CMVT_v1 was a gift from David Sabatini [Addgene plasmid #133072]) expression plasmid with a N-terminal FLAG tag. Mutant and truncated VCPIP1 constructs were generated by excising and replacing sequences using primers with PCR, T4 polynucleotide kinase ligation (mixture of DNA, T4 ligation buffer, PNK (NEB M0201S), T4 DNA ligase (NEB, M0202S) and H₂0) and 1 hincubation at 37 °C[68].

All VCPIP1 variants and VCP were expressed transiently in Expi293 (Thermo Fisher Scientific, A14635) following the manufacturer's manual. Cells were harvested 60-65 h post-transfection and lysed by sonication in lysis buffer (50 mM HEPES pH 7.4, 200 mM NaCl, 5% glycerol) supplemented with protease inhibitors. The lysate was cleared by ultracentrifugation (176,590 x g, 45 min) and then incubated with either StrepTactin®XT 4Flow high-capacity resin (IBA life sciences, 2-5030-002) or FLAG-antibody-coated beads (Genscript, L00432). Bound proteins were eluted with 50 mM of biotin or 0.2 mg/mL 1xFLAG (DYKDDDDK) peptide. Proteins were further purified by ion exchange chromatography (IEX) with a Poros 50HQ (Thermo Fisher Scientific, 1255911) column, eluting with a linear NaCl gradient from 200−750 mM. Elution fractions were concentrated and polished by size exclusion chromatography (SEC) in SEC buffer (30 mM HEPES pH

7.4, 150 mM NaCl) using Superose6 Increase 10/300 GL (VCP alone) and Superdex200Inc 10/300 GL (VCPIP1 alone) columns (Cytiva). For purification of co-expressed VCP-VCPIP1 complex, ion exchange chromatography was omitted.

p47 was cloned into pNIC-Bio2 with an N-terminal His-6-TEV tag. For expression, LOBSTR *E. coli* expression strains (Kerafast, EC1002) were transformed, and a 1 L culture was grown in TB at 37 °C to OD₆₀₀ = 0.6, and induced with 0.5 mM Isopropyl β-D-1-thiogalactopyranoside (IPTG). After induction, temperature was decreased to 18 °C and the protein was expressed overnight. The culture was centrifuged (1968 × g, 20 min), resuspended in lysis buffer (50 mM HEPES pH 7.4, 200 mM NaCl, 20 mM imidazole, 5% glycerol, benzonase and protease inhibitors), lysed using sonication, and the lysate was cleared using ultracentrifugation (176,590 × g, 60 min). Clarified lysate was applied to high affinity Ni-charged resin (Genscript, L00223) and eluted with increasing imidazole concentrations (20-750 mM). p47 was further purified using ion exchange chromatography – the fractions were diluted to 50 mM NaCl, applied to a PorosHQ and eluted with a NaCl gradient from 50-750 mM. Peak fractions were concentrated using centrifugal concentrators (30 kDa MWCO, Millipore Amicon, UFC3090) and applied to a Superdex75 Increase 10/300 GL (Cytiva) equilibrated in SEC buffer (30 mM HEPES pH 7.4, 150 mM NaCl). Final protein samples of all constructs were flash-frozen in liquid nitrogen and stored in −80 °C.

### EM sample preparation and data collection

For data set 1 (VCP-VCPIP1): VCP-VCPIP1 was crosslinked with a 400x molar excess of bis(sulfosuccinimidyl)suberate (BS3) crosslinker (9.8 μM BS3 (12 mM), 0.5 mg VCP-VCPIP1 sample (5.2 mg/mL) and SEC buffer), quenched with 100 mM Tris and loaded on Superose6 Increase 10/300 GL (Cytiva) which was equilibrated with SEC buffer for further purification. Final concentrated sample (2.4 mg/mL) was diluted to 1.8 mg/mL and mixed with CHAPSO (0.2 mM final concentration) directly prior to grid preparation. 4 μL of sample were applied to the grid and blotted for 3 s followed by a 3 s post-blot incubation before vitrification.

For data set 2 (VCP-VCPIP1-p47): Individually purified VCPIP1, VCP and p47 were mixed and incubated for 30 min on ice, loaded on Superose6 Increase 10/300 GL (Cytiva) and concentrated (30 kDa MWCO) to 4.7 mg/mL. Sample was diluted to 2.0 mg/mL and CHAPSO (0.2 mM final) was added directly prior to grid preparation. 4 μL of sample was applied once to the grid and blotted for 3 s followed by a 3 s post-blot incubation before vitrification.

Quantifoil 1.2/1.3 300, N1-C14Cu30-50 grids were glow discharged for both data sets at 20 mA, 60 s, 39 Pa. Subsequently, grids were vitrified in a Leica EM-GP operated at 10 °C and 90% relative humidity with 10 s pre-blot time.

Data Set 1 was collected using SerialEM[69] (v3.8.6) in a Thermo Scientific Titan Krios equipped with a Gatan Quantum image filter (20 eV slit width) and a post-GIF Gatan K3 direct electron detector. Movies were acquired at 300 kV at a nominal magnification of 105,000x in counting mode. 11,590 movies (50 frames each) were recorded with 3 exposures per hole and 9 holes per stage position resulting in 27 image acquisition groups. Defocus was varied from −0.9 - −2.2 μm and total dose, pixel size and exposure time were 53.4 e−/ Å², 0.83 Å and 2.83 s, respectively.

Data Set 2 was collected using EPU (v3.7) in a Thermo Scientific Titan Krios equipped with a Selectrix energy filter (10 eV slit width) and a post-GIF Falcon4 direct electron detector. Movies were acquired at 300 kV at a nominal magnification of 165,000x in counting mode. 12,261 movies (49 frames each) were recorded with (3) per hole. Holes per stage position were calculated by EPU software. Defocus was varied from −0.8 - −2.2 μm and total dose, pixel size and exposure time were 49.2 e−/ Å², 0.73 Å and 2.87 s, respectively.

## Data processing and model building

Initial processing was done with cryoSPARC[70] (v3.3.1) and later processing and refinement was done with cryoSPARC (v4.6.0). Resolutions are stated based on the Fourier shell correlation (FSC) 0.143 threshold criterion[71].

Data set 1 (VCP-VCPIP1): 11,580 movies were corrected for beam induced motion and CTF was estimated on-the-fly using cryoSPARC live. A total of 2,050,270 particles were extracted (1.13 Å/pix) after crYOLO[72] particle picking from 9365 curated micrographs. The consensus structure was resolved at 2.3 Å using homogenous refinement after two rounds of heterogeneous refinement. 3D variability analysis was performed on these particles (1,369,446), followed by homogenous refinement, leading to a 2.9 Å reconstruction (107,859 particles) of three VCPID regions binding to the C-terminal end of VCP. To resolve more density for one VCPID, C6 symmetry expansion was done on the particles from the consensus refinement resulting in 8,216,676 particles followed by local clustering using 3D variability. A 2.9 Å reconstruction was resolved after local refinement of 592,039 particles. Lastly, the symmetry expanded particles were downsampled (2.6 Å/pix) followed by local clustering with 3D variability, 3D classification, 3D variability, and local refinement to resolve density for VCPIP1 UBX domain at a resolution of 3.1 Å. Final particle stacks were polished using reference based motion correction (restoring the particles to 0.83 Å/pix) and final refinements were performed with per-particle defocus estimation and correction of higher order CTF aberrations per-acquisition group (Supplementary Figs. 2, 3). All final maps were postprocessed with deepEMhancer[73], and these maps, together with the sharpened and unsharpened maps from cryoSPARC were used for model building. Initial models for VCPIP1 consensus, 3xVCPID, VCPID, and VCPIP1 UBX domain were generated by rigid-body docking (in ChimeraX (v1.6)[74]) in a combination of VCP from the Protein Data Bank (PDB) (PDB:5FTK[16]) with truncations of the AlphaFold model for VCPIP1 (UBX domain, VCPID and stalk region) (AF-Q96JH7-F1-v4). The combined models were first flexibly fit using ISOLDE[75], followed by residue-by-residue modeling and inspection with COOT (v0.9.8)[76] and then finally refined using phenix.real_space_refine[77] using reference restraints, after preparation using phenix.ready_set[78].

Data set 2 (VCP-VCPIP1-p47): 12,261 movies were corrected for beam induced motion and CTF was estimated on-the-fly using cryoSPARC live. A total of 2,003,371 particles were extracted (at 1.96 Å/pix) from 12,129 curated micrographs after cryoSPARC template picking. Two rounds of heterogenous refinement were done initially to classify out a VCP dodecamer class. Two subsequent rounds of heterogenous refinement were done to classify out VCP states. From the fifth heterogenous refinement run, one class that had obvious VCPID density was used to resolve the 3xVCPID structure of three VCPID binding to the C-terminal end of VCP. For the 3xVCPID structure, homogeneous refinement was done after heterogenous refinement and further classification was carried out with 3D variability. Two classes from the 10 classes of 3D variability were selected for local refinement, followed by two rounds of heterogenous refinement. Lastly, local refinement was done resulting in a 3.1 Å reconstruction from 77,321 particles. For the p47 UBX domain structure, particles from heterogeneous refinement were combined and C6 symmetry expansion was applied, resulting in 4,293,558 particles. Local clustering with 3D variability was carried out with a mask on the UBX region, followed by 3D classification, another round of 3D variability, local refinement, back-to-back rounds of 3D variability and a final local refinement, resulting in a 4.0 Å reconstruction (Supplementary Figs. 6, 7). The model from data set 1 was used as a starting point for model building, and AlphaFold3[54] was used to predict the p47 linker, the SHP motif and the SEP domain interaction with VCP. The highest rank prediction from AlphaFold3 of VCP N and D1 domains and full length p47 was used as the initial model for p47 density bound to VCP N-domain. This model was first flexibly fit using target restraints in ISOLDE, refined using phenix.real_space_refine with

reference restraints, and finally inspected with COOT. Local resolution ranges are given based on 0–75% percentile in local resolution histograms[79]. Structural biology applications used in this study were configured by SBGrid[80].

## Interface Residues

Initial identification of interacting residues between VCP and VCPIP1 was determined based on visual inspection of density in parallel to scores corresponding to interaction using the script: residue_energy_breakdown_script.sh. We also used PDBePISA[81] to confirm residues with the highest buried area percentage. Residue predictions for p47 and VCP N-domain interaction are based on the AlphaFold3 model and were similarly cross-validated using PDBePISA and the script.

## Fluorescent protein labeling for TR-FRET assays

VCPIP1 constructs (VCPIP1$^{WT}$, VCPIP1$^{\Delta VCPID}$, VCPIP1$^{\Delta UBX}$, VCPIP1$^{\Delta VCPID \, \Delta UBX}$) and p47 were non-site selective labeled with BODIPY and VCP was labeled with Terbium (Tb) (CoraFluor: R&D systems, Cat. No. 7920) in a 1:1 molar ratio as previously described[82]. The samples were incubated for 1 h at room temperature. After incubation, the samples were quenched with 20 mM Tris pH 8, incubated for 10 min and spun in a Zeba Spin Desalting Columns (Thermo Fisher Scientific, 89882) to remove excess BODIPY and Tb and buffer exchange back into SEC buffer. A280 and A503 readings were taken to calculate degree of labeling (DOL) for BODIPY labeled proteins; VCPIP1$^{WT}$ (0.23), VCPIP1$^{\Delta VCPID}$ (DOL = 0.62), VCPIP1$^{\Delta UBX}$ (DOL = 0.54), VCPIP1$^{\Delta VCPID \, \Delta UBX}$ (DOL = 0.44), p47 (DOL = 0.27). The extinction coefficient of 80,000 M$^{-1}$cm$^{-1}$ was used for BODIPY at 503 nm. A280 and A340 readings were taken to calculate the DOL of Tb-VCP (DOL = 0.33). The extinction coefficient at A$_{340}$ of CoraFluor was equal to 22,000 M$^{-1}$cm$^{-1}$ as described[82].

## TR-FRET binding and competition assays

For binding assays, Tb-VCP was mixed with TR-FRET buffer (25 mM HEPES pH 7.4, 150 mM NaCl, 0.1% BSA, 0.05% Tween-20). A volume of 7.5 µL of the mixture was dispensed into each well of a 384-well plate (Corning, 4514). The serial dilution of labelled protein was performed in 96-well plate, and 7.5 µl of the serial dilution mixture was added to each well of the 384-well plate. The final assay volume per well was 15 µL. The plate was incubated for 1 h at room temperature. Fluorescence signals were measured using a PheraStar FS plate reader (BMG Labtech). Tb was excited at 337 nm and emitted at 490 nm (Tb) and 520 nm (BODIPY) which were recorded with a 70 µs delay over a window of 600 µs. The plates were measured for 6 cycles. TR-FRET ratios are calculated as a ratio of signal detected at 520/490 nm. All assays were performed as technical triplicates. Data analysis was done using nonlinear regression in GraphPad Prism (v10.2.3).

For binding assays with ATP analogs, serial dilution of 1 nM labeled BODIPY VCPIP1 WT protein was prepared in TR-FRET buffer with 1 uM of ATP (Sigma Aldrich, A7699-10G), ADP (Sigma Aldrich, A2754), AMP (Sigma Aldrich, A1752) or ATPγS (Jena Biosciences, NU-406-50). All other assay conditions were performed the same as the binding assays described above.

For displacement assays, demonstrating the displacement of BODIPY-VCPIP1$^{WT}$ by unlabeled-VCPIP1$^{WT}$ or p47 from Tb-VCP, 10 nM BODIPY-VCPIP1$^{WT}$ was incubated with 10 nM Tb-VCP. Serial dilution of unlabeled proteins (VCPIP1$^{WT}$ or p47) was prepared in a 96-well plate. All other conditions to the assay were performed the same as the binding assays described above.

## Ubiquitin rhodamine assay

For Ub-Rho activity assays, enzyme concentration varied while substrate (Ub-Rho) concentration remained constant. Each VCPIP1 construct (VCPIP1$^{WT}$, VCPIP1$^{\Delta VCPID}$, VCPIP1$^{\Delta UBX}$, VCPIP1$^{\Delta VCPID \, \Delta UBX}$) was tested with and without VCP. VCP was also tested alone as a control. Enzyme

**Table 1 | Total number of Golgi quantified per condition**

| Condition | Total # of Golgi quantified |
|---|---|
| WT | 344 |
| WT + VCPIP1$^{EmptyVector}$ | 318 |
| KO | 356 |
| KO + VCPIP1$^{EmptyVector}$ | 341 |
| KO + VCPIP1$^{WT}$ | 338 |
| KO + VCPIP1$^{\Delta VCPID}$ | 340 |
| KO + VCPIP1$^{\Delta UBX}$ | 364 |
| KO + VCPIP1$^{\Delta VCPID\Delta UBX}$ | 353 |
| KO + VCPIP1$^{C219A}$ | 359 |

dilutions were prepared in assay buffer (50 mM Tris pH 8.0, 0.5 mM EDTA, 5 mM TCEP, 11 μM ovalbumin), for final concentracon ranging from 500 nM to 7.8 nM. 10 μL of enzyme dilutions were distributed in a 384-well plate (Corning, 3820). Ub-Rho solution was prepared in the same assay buffer, for a final concentration of 500 nM. 10 μL of Ub-Rho solution was transferred to the 384-well plate and quickly centrifuged before reading on the CLARIOstar (BMG Labtech). Fluorescence was measured for 1 h, every 30 s at an excitation and emission of 487 nm and 535 nm, respectively. Fluorescence over time was plotted and initial velocities analyzed using simple linear regression in GraphPad Prism. All assays were performed in duplicate.

## CRISPR CAS-9 KO of VCPIP1

**Cell Culture.** A2058 (a gift of Peter Sorger's lab) and HEK-293T (ATCC catalog # CRL-3216), were cultured and maintained in DMEM (Gibco, 11965092) supplemented with 10% (v/v) fetal bovine serum (Sigma, F0926) and 100 U/mL penicillin + 0.1 mg/ml streptomycin (Gibco, 15070063). Cells were grown and maintained in tissue-culture treated dishes (Corning, CLS430167) at 37 °C with 5% $CO_2$ in a water-jacketed $CO_2$ incubator. For passaging, cells were washed with PBS (Gibco, 10010023) detached using 0.25% trypsin/EDTA (Gibco 25200056) for passaging. All cell lines were routinely tested for mycoplasma and verified to be mycoplasma free (Lonza, LT07-703).

## Cas9 Expressing Cell Lines

A2058 were transduced with Cas9-Flag via lentiviral infection (Cas9-Flag plasmid was a gift of Peter Sorger's lab). Following blasticidin selection (Gibco, A1113903), a sample of cells were harvested, pelleted by centrifugation (5000 × g for 10 min at 4 °C), lysed on ice with lysis buffer (20 mM Tris pH 8, 150 mM NaCl, 1% NP-40, 10% glycerol, 1 mM TCEP and protease inhibitor cocktail (Thermo Scientific, 78429) and clarified by centrifugation after 30 min. Protein content was quantified by BCA (Thermo Scientific, 23225). Lysate was diluted to 2 mg/mL in lysis buffer. 4x LDS sample buffer (Thermo Fisher, B0008) supplemented with 10% BME was added to each sample. Following mixing, samples were heated to 95 °C for 10 min. Then samples were resolved by SDS-PAGE and analyzed by Western blot to confirm FLAG expression (Invitrogen, MA1-91878) with GAPDH control (Cell Signaling Technology, 2118).

## Creation of stable VCPIP1 KO Cell Line

A2058 VCPIP1 KO cells were generated using the Dharmacon Edit-R crRNA system. Lyophilized crRNA (Horizon Discovery; VCPIP1 #1 CM-019137-01-0002; #2 CM-019137-02-0002; CM-019137-03-0002; #4 CM-019137-04-0002; NTC #1) and tracrRNA (Horizon Discovery, U-002005-05) were resuspended in buffer (Horizon Discovery, B-006000-100) to a final concentration of 10 uM per manufacturers protocol. A2098-Cas9-Flag cells (40,000 cells) in media without antibiotic were seeded in 12-well plates (Corning, 3513) and allowed to adhere overnight. Mixtures of tracrRNA and all 4 VCPIP1 guides,

tracrRNA and NTC guide, and Dharmafect 1 (Horizon Discovery, T-2001-01) were prepared in Opti-MEM (Gibco, 31985070). Dharmafect mixture was added in equal volumes to the tracrRNA-VCPIP1 and tracrRNA-NTC solutions. After 20-minute incubation, the mixtures were added to appropriate wells for a final concentration of 2 uL/mL Dharmafect, 25 nM tracrRNA, and 25 nM VCPIP1 (6.25 nM of each guide) or 25 nM NTC. Cells were incubated overnight at 37 °C. Media was aspirated and replaced after 18 h. Cells were permitted to grow to confluency for 2 days, upsized to 6-well plates (Corning, 3516). A sample of cells were harvested, lysed, and resolved by SDS-PAGE and analyzed by Western Blot as above to assess VCPIP1 knock-out (Bethyl, A302-933A-M) with GAPDH control (Cell Signaling Technology, 97166). Monoclonal VCPIP1 KO cells were obtained through a limiting dilution plating 0.5 cells/well in a 96-well plate (Corning, 3598), with monoclonal populations being upsized to 24-well plates (Corning, 3524), with samples being taken when confluent to identify complete VCPIP1 knock out populations.

## Golgi staining and imaging

A2058 cells, WT and KO, were grown to 70–80% confluency and then split (DMEM + 10% FBS) directly into 8-well ibidi slides (ibidi, 80807-90) to 0.05 × 10$^6$ density in 0.2 mL per well. Cells were transfected the next day with a mixture of 16 μL of Opti-MEM (Gibco, 31985-070), 0.94 μL FuGENE ® HD Transfection Reagent (Promega, E2311) and 0.31 ug of DNA in a drop-by-drop manner. Each slide was plated with the first eight conditions with the ninth condition plated on a separate slide. The cells were incubated for 24 h post transfection, followed by a 15-minute fixation with 4% PFA at RT. Next, cells were washed with PBS and blocked using blocking buffer (PBS, 3% BSA, 0.03% Triton-X 100, 5% Goat Serum) for 1 h at RT before the cells were incubated overnight at 4 °C with GM130 antibody (D6B1) XP Rabbit mAB (Cell Signaling Technology #12480) (1:200). After three washes with PBS (5 min each at RT), goat-anti rabbit secondary (Invitrogen, #A27040) (1:1000) was added and incubated at RT for 1 h before three more PBS washes. The stained cells were mounted with Prolong Glass with NucBlue and left to dry uncovered in a dark, humid environment for 24 h at RT and stored at 4 °C until the conditions were analyzed using confocal microscopy.

The mounted slides were analyzed (ZEN Desktop software, (RRID:SCR_013672)) using a Zeiss 980 confocal microscope with an Airyscan 2 detector enabling super-resolution and high-speed imaging. We collected images of all the conditions using the 63 × 1.4 oil objective. Airyscan processed images were imported into Fiji[83] (ImageJ). Channels from the final image were split to see both DAPI and GM130 antibody detection independently and then merged. All images were imaged and processed with the same parameters.

For western blot analysis, 10$^6$ cells were grown on a 12-well plate (Corning, 3513), transfected, trypsinized, lysed in RIPA Lysis and extraction buffer (G Biosciences, 768-489) with cOmplete Mini protease inhibitor cocktail tablets (Roche, 11836153001), with 1 uL of Benzonase (Sigma Aldrich, 70664-3) and the supernatant were transferred onto PVDF membranes using an iBlot 2 dry blotting system (Thermo Fisher Scientific, IB21001). Rabbit anti-VCPIP1 primary antibody (BETHYL, A302-933A) was used to detect the VCPIP1 in WT and KO conditions and the blots were imaged using LI-COR Odessey CLx detecting a donkey anti-rabbit secondary antibody (LI-COR, 926-32213). The blot demonstrated that the transfection of all the constructs was successful within the A2058 KO cell line (Fig. 4a).

## Quantification

A total of three biological replicates with two technical replicates each were imaged for the nine conditions, where each technical replicate was imaged on a separate day (six total days) with the same imaging settings. For visualization and quantification, images were imported into the imaging tool ImageJ's Fiji and brightness/contrast were chosen for a single image and set for the remaining images. Images were

analyzed in the reverse order they opened (last image was analyzed first). Next, a freeform selection was drawn around each Golgi that was fully visible in the field of view and the area of the freeform selection was calculated in Fiji by selecting the area parameter in the Set Measurement tool and then using the Measure tool to calculate each Golgi selection. Areas of ~55 unique Golgi per replicate per condition were measured. Images in each replicate were analyzed and measurement concluded once ~55 Golgi per replicate per condition were quantified. The total number of Golgi quantified per condition is represented in Table 1.

### Reporting summary

Further information on research design is available in the Nature Portfolio Reporting Summary linked to this article.

## Data availability

Cryo-EM maps and models will be available from Electron Microscopy Data Bank (EMDB) and RCSB Protein Data Bank (PDB) under the accession code numbers: VCP-VCPIP1 complex (https://www.ebi.ac.uk/pdbe/entry/emdb/EMD48503), VCP consensus (EMD-48499 (https://www.ebi.ac.uk/pdbe/entry/emdb/EMD48499], PDB-9MPQ (https://doi.org/10.2210/pdb9MPQ/pdb), 3xVCPID (EMD-48500 (https://www.ebi.ac.uk/pdbe/entry/emdb/EMD48500), PDB-9MPR (https://doi.org/10.2210/pdb9MPR/pdb), VCPID (EMD-48501 (https://www.ebi.ac.uk/pdbe/entry/emdb/EMD48501), PDB-9MPS (https://doi.org/10.2210/pdb9MPS/pdb) and UBX (EMD-48502 (https://www.ebi.ac.uk/pdbe/entry/emdb/EMD48502), PDB-9MPT (https://doi.org/10.2210/pdb9MPT/pdb) and VCP-VCPIP1-p47 complex (EMD-48504 (https://www.ebi.ac.uk/pdbe/entry/emdb/EMD48504), 3xVCPID (EMD-48506 (https://www.ebi.ac.uk/pdbe/entry/emdb/EMD48506), PDB-9MPV (https://doi.org/10.2210/pdb9MPV/pdb) and p47 (EMD-48505 (https://www.ebi.ac.uk/pdbe/entry/emdb/EMD48505), PDB-9MPU (https://doi.org/10.2210/pdb9MPU/pdb). The EMDB and PDB codes are listed in Supplementary Table 1 and Table 2. The source data underlying Figs. 2, 4 and Supplementary figs. 4, 5, 8 are provided in the Source Data file. Source data are provided in this paper. Source data are provided with this paper.

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

## Acknowledgements

This work was supported by NIH NCI 5F31CA281197-02 (to B.S.), NIH NCI RO1CA262188 (to E.S.F.) and NIH NCI RO1CA233800 (to S.J.B.). We thank the staff at the Harvard Cryo-EM Center for Structural Biology for their outstanding support during grid screening and data collection and the staff at the Dana Farber Cancer Institute Molecular Imaging Core (MIC) for their microscopy training and support. We acknowledge SBGrid for assistance with software and high-performance computing[80]. We acknowledge Milka Kostic for her valuable input and critical feedback on the manuscript and members of the Buhrlage and Fischer labs for discussions, help and feedback.

## Author contributions

B.S., M.H., S.J.B., and E.S.F. conceived study and designed research plan. B.S. cloned and purified proteins, performed biochemical assays and cellular assays and, with the support of M.H., conducted cryo-EM structure determination. A.B. created the VCPIP1 KO cell line. H.Y. and I.J.M. provided technical insight and initial analysis for biochemical assays. B.S., M.H., S.J.B., and E.S.F. designed experiments, and B.S. analyzed and interpreted data. S.J.B. and E.S.F. supervised the study and acquired funding. B.S. prepared figures and wrote the original manuscript. B.S., M.H., S.J.B., and E.S.F. reviewed and revised the manuscript. All authors approved of the final version of the manuscript.

## Competing interests

E.S.F. is a founder, scientific advisory board (SAB) member, and equity holder of Civetta Therapeutics, Proximity Therapeutics, Stelexis Biosciences, Anvia Therapeutics (also board of directors) and Neomorph, Inc. (also board of directors). He is an equity holder and SAB member for Avilar Therapeutics, Photys Therapeutics, and Ajax Therapeutics and an equity holder in Lighthorse Therapeutics and CPD4, Inc. (also board of directors). E.S.F. is a consultant to Novartis, EcoR1 Capital, Odyssey and Deerfield. The Fischer lab receives or has received research funding from Deerfield, Novartis, Ajax, Interline, Bayer and Astellas. S.J.B. is a founder, SAB member, and equity holder of Entact Bio and receives or has received sponsored research funding from Novartis Institutes for Biomedical Research, AbbVie, Kinogen, TUO Therapeutics, Takeda, and Pivotal Life Sciences. The remaining authors declare no competing interests.
