## [Transparent Peer Review file · Nature Communications]

Structural basis of VCP-VCPIP1-p47 ternary complex in Golgi maintenance

Corresponding Author: Professor Eric S. Fischer

Version 0:

Reviewer comments:

Reviewer #1

(Remarks to the Author)

Cellular functions of the hexameric ATPase VCP/p97/Cdc48 are controlled by a variety of regulatory cofactor proteins. Shah et al. solved the three-dimensional structure of VCP in complex with the deubiquitinase (DUB) cofactor VCPIP1 (also known as VCIP135) by cryo-EM, both in the absence and presence of an additional cofactor, p47. Previous work has shown that all three proteins cooperate in the post-mitotic reassembly of Golgi stacks. The VCP-VCPIP1 complex has a 6:3 stoichiometry, with one VCPIP1 protomer binding to two adjacent VCP protomers. Binding is mediated via two major interfaces: one is formed between VCPIP1's UBX domain and the VCP N-domain, and the other between the newly assigned, central VCPID domain of VCPIP1 and the bottom of VCP's D2 domain. Densities for VCPIP1's catalytic OTU domain and a stalk domain linking it to the VCPID domain are visible but were of insufficient quality for model building. Next, the authors determined the contributions of the UBX and VCPID domains to VCP binding and DUB activity, respectively, in biochemical assays. They found that the UBX domain was required for VCP binding, but not DUB activity, and vice versa for the VCPID domain. They also found that the DUB activity was stimulated by VCP binding. The structure of the ternary VCP-VCPIP1-p47 complex is very similar to the VCP-VCPIP1 complex, except that additional densities at VCP's N-domains indicated the binding of three UBX domains from p47 instead of two UBX domains from VCPIP1, suggesting competition of the two cofactors for binding to the N-domains. A binding competition experiment showed that p47 binds with about 3.5-fold higher affinity than VCPIP1 to VCP, which was attributed to the additional interaction between p47's SHP box and VCP's N-domain by the authors. To test the relevance of the VCPID and UBX domains for a known cellular function of VCPIP1, the authors complemented a VCPIP1 knockout cell line with plasmids encoding wild-type or mutant VCPIP1 and stained the Golgi apparatus. They concluded that the VCPID and UBX domains are both required for efficient Golgi reassembly.

The structural analysis of VCP-cofactor complexes is of high interest both from a basic research and biomedical perspective. Unfortunately, the present study of VCP-VCPIP1 complexes is compromised by its limited novelty compared to two recent papers by Liao et al (ref. 41, published) and Vostal et al (ref 40, preprint server). The overall structure of the VCP-VCPIP1 complex, the new binding interface between VCPID and the D2 domain of VCP, its importance for Golgi reassembly in cellulose, and the VCP-stimulated DUB activity of VCPIP1 were all reported in one or both of the competing studies. Vostal et al additionally solved the OTU domains of the VCP-VCPIP1 complex and proposed a model for the de-ubiquitination of substrate proteins at the VCP exit site. The structure of the ternary VCP-VCPIP1-p47 complex, reported for the first time in the present manuscript, provides only incremental new insight compared to the previous binary VCP-VCPIP1 and VCP-p47 structures. Moreover, parts of the structure-function analysis are incomplete, and the imaging data related to Golgi fragmentation are unconvincing.

Specific points:

1. Fig. 2: There is clearly residual VCP binding for both VCPIP1 variants lacking the UBX domain, likely derived from interactions of the VCPID domain (cf. Liao et al, Vostal et al) and/or the SHP box recently identified by Nakayama & Kondo (res. 1018-1036; see ref. 32) - not shown in Fig. 1a. The authors should either quantify the residual binding using higher protein concentrations, or re-phrase their conclusions. Moreover, including the DUB activity of the isolated OTU domain could support the interpretation of the presented DUB data.
2. Fig. 3, Results lines 215ff.: The discussion of the competitive/simultaneous binding of VCPIP1 and p47 to VCP is incomplete. First, can the authors exclude the possibility that the three p47 UBX domains occupy N-domains spared by the

VCPIP1 UBX domains? Second, if deletion of its UBX domain strongly decreases VCPIP1's affinity for VCP (Fig. 1), how can the ternary complex be stable in the absence of its UBX domain binding to VCP's N-domain? Third, the authors propose that the competition by/ higher affinity of p47 can be explained by additional contacts via its SHP box. Please note that VCPIP1 also has a SHP box (see above) that was shown to bind VCP (refs. 32, 41), and that p47 has a second SHP box (ref. 39), complicating the interpretation of competitive/simultaneous binding. More detailed insights would require the comparative analysis of mutations in all three SHP boxes.

3. Fig. 4: The Golgi staining in the different cells is not convincing at all. Much clearer images of Golgi fragmentation can be obtained by immunofluorescence staining using a GM130 antibody (see e.g. refs. 32, 41). The VCPIP1 WT and KO cells should be transfected with empty vector for a proper control. Quantification of Golgi fragmentation in at least three biological replicates is mandatory.

4. Introduction, lines 39/40: Hemmo Meyer's work on ubiquitin-independent PP1 maturation by VCP-p37 should be mentioned.

5. Results, line 218: The role of the SHP box in VCP binding was first reported in refs. 24 and 33.

6. Discussion, lines 291ff.: Please explain in more detail how the new structure "helps to rationalize effects of CB-5083". Please note that clinical trials of CB-5083 and CB-5339 had to be discontinued.

Reviewer #2

(Remarks to the Author)

In their study Shah et al. investigate ternary complex formation of the three proteins VCP, VCPIP1 and p47. The main conclusions are drawn from cryo-EM data. I will focus here on the time-resolved FRET (TR-FRET) and fluorescence data. The authors use trFRET to determine the K_D 's of several truncation mutants and to conclude that the UBX domain anchors VCPIP1 to VCP (Fig. 2). In principle I support their conclusion, but there remain a few questions on the experiments (see below). In addition, the authors conclude from the images in Fig. 4 that VCPIP1 and VCP interaction is necessary for Golgi reassembly. Here I cannot fully follow the argumentation. These are my specific questions:

- 1) Nomenclature: Why do the authors call the FRET-based method 'time-resolved'? I do not see any time-resolved data. If there is some, please show that.
- 2) Please define what the TR-FRET ratio is (i.e. the y-axis in Fig. 2). I even could not find this information in the main text of reference 60.
- 3) Why does the TR-FRET ratio reach only about 0.3 for VCPIP1(WT) and almost 1.0 for VCPIP1(deltaVCPID), although both have about the same affinity?
- 4) Do the authors have any idea, why the DOL is so different (low) for VCPIP1(WT) compared to e.g. VCPIP1(deltaVCPID)? I think that the BODIPY can react with any lysine and therefore should bind almost in a stoichiometric way?
- 5) Fig. 4: Why were Z-stack max intensity projections used?
- 6) Fig. 4: The white arrows are supposed to indicate "gaps" in the cytoplasm, the red ones "maximal intensity spots". For both, I cannot see a difference between VCPIP1(WT) and the other cells in the cytoplasm. Could you please quantify what a "gap" and what a "maximal intensity spot" is and then give an estimate on how much a "more pronounced Golgi fragmentation" is?
- 7) What should I learn from the merged images in Fig. 4?

Reviewer #3

(Remarks to the Author)

The manuscript reports work focused on the characterisation of VCP complexes involved in Golgi reassembly (VCP-VCPIP1 and VCP-VCPIP1-p47) using single particle cryo-EM and evaluating the functionality of the VCP-VCPIP1 interaction in cells.

Strengths of the manuscript include the detailed characterization of the mode of interaction between VCPIP1 and the D2 and N domains. This work represents a tour de force in image analysis, successfully extracting high-resolution information from highly heterogeneous complexes, likely affected by both conformational and compositional variability. The study also demonstrates that the UBX domain of VCPIP1 is competitively displaced by the UBX domain of p47. It further defines the detailed interaction between the p47 UBX-SHP linker and the N-domain of VCP. The cellular studies suggest that all domains of VCPIP1 play a critical role in Golgi reassembly.

Despite providing valuable insights, the manuscript comes across as rushed, with certain sections appearing incomplete and some conclusions lacking accuracy. The reason for this urgency becomes evident in the discussion section, where the authors mention two other manuscripts addressing similar work on the VCP-VCPIP1 complex. One is still in preparation (submitted to Bioarchives), while the other was published in *Advanced Science* in September 2024, shortly before this manuscript was submitted. While the pressure of such circumstances is understandable, it does not excuse the incomplete analysis of the work conducted or the failure to fully leverage its potential. The manuscript faces challenges on two fronts: stylistic weaknesses and design/quality issues. Below, I will outline these shortcomings in detail.

Starting with the introduction, the authors describe the role of VCP in the ubiquitin-proteasome system (UPS), which feels unnecessary given the broader relevance of the topic to VCP's role in Golgi assembly. Moreover, the description is heavily

biased, disproportionately emphasizing work published by Harvard groups while neglecting significant contributions from other researchers. It also conflates findings on VCP with those of its yeast homologs, leading to confusion and a lack of clarity.

Regarding the analysis of the VCPIP1-VCP complex, the authors provide a detailed and well-described characterization of the interaction between the VCPIP1-UBX domain and VCP, as well as another domain referred to as VCPIP1-VCPIID interacting with VCP. However, greater care should be taken when drawing conclusions about the 1:2 stoichiometry (VCPIP1:VCP), particularly given the extensive data excluded during the classification analysis, as shown in Extended Data Fig. 2. The conclusions are based on a subset of 233,953 particles, corresponding to only 38,992 particles considering the symmetry extension for the UBX domain and just 17,976 particles for VCPIID- amounting to less than 3% of the particles analyzed.

The data appears to contain significantly more information than the authors choose to present or analyze. Notably, some of the classes clearly show densities above the D1 rings, potentially reflecting the upward motion of the N domain. However, this valuable information was lost in the consensus reconstruction, where all N domains are depicted as coplanar with the D1 ring. Similarly, this information is absent in the final D1-UBX density derived from less than 3% of the data. The resolved density of VCPIP1 interacting with the D2 domain is minimal in this study compared to the other two manuscripts mentioned above on VCPIP1. The provided movie is unconvincing in demonstrating the movement of these regions of VCPIP1; instead, it appears to reflect particle occupancy rather than dynamic motion. Have the authors considered performing focused classification specifically on VCPIP1, excluding VCP densities?

The subtitle referring to the VCP-VCPIP1-p47 complex is inaccurate, considering how little of the VCPIP1 and p47 proteins are resolved in the study. Similarly, the title "VCPIP1 catalytic activity and binding to VCP are required for Golgi reassembly" is also misleading, as the analysis primarily tested the effects of mutants lacking specific domains of VCPIP1, rather than mutants specifically targeting the interaction surfaces defined by the single-particle cryo-EM work.

The Golgi reassembly data presented using the full-length and domain deletion mutants is of overall poor quality, primarily due to its low resolution. This makes it difficult to distinguish between Golgi fragmentation (observed as speckles in the images, which are also present in VCPIP1 knockout cells with VCPIP1 reintroduced) and abnormal nuclear morphology. No quantification of the data has been attempted, and it is difficult to believe that the authors did not have access to higher-resolution light microscopes to more clearly demonstrate defects in Golgi stack reassembly.

It is surprising that the authors did not utilize higher-resolution light microscopes to more clearly demonstrate defects in Golgi stack reassembly, nor did they attempt any quantification of the data.

Finally, and perhaps most importantly, no comparison has been made between the data presented in this manuscript and the findings of the manuscript published by Lao and colleagues in *Advanced Science* (reference 41). The only comment provided is that the other manuscript describes "some of the features... and further supports our conclusions," which is insufficient for a meaningful analysis or discussion of overlapping results. There are also clear differences between these two studies, particularly in the binding of VCPIP1-UBX to VCP-N and the conformation adopted by the VCP N domain upon binding. These discrepancies warrant further discussion and analysis, as they could provide valuable insights into the mechanistic differences observed. Especially noteworthy is the fact that the two studies used different expression systems: mammalian cells in this study and bacteria in the *Advanced Science* manuscript. Interestingly, the nucleotide occupancy of VCP has been shown to differ depending on the expression system used. Specifically, VCP is nucleotide-free when expressed in mammalian cells, whereas it predominantly binds ADP when expressed in bacteria. This difference could significantly influence the observed interactions and conformations, further emphasizing the need for a detailed comparison between the two studies. The authors already have access to the necessary proteins and a beautiful TR-FRET assay, which would enable them to address these questions effectively and provide more comprehensive insights.

Version 1:

Reviewer comments:

Reviewer #1

(Remarks to the Author)

The authors have satisfactorily addressed my concerns. The revised manuscript is suitable for publication in NCOMMS.

Reviewer #2

(Remarks to the Author)

The authors have addressed my comments. In particular, they have now explained TR-FRET and re-designed the cellular fluorescence experiments.

Nevertheless, there remains some ambiguity concerning the quantification of the area of Golgi (Fig. 4). I could not find a clear description on the quantification. When I look at the images provided e.g. in Fig. 4c, I first notice that the contrast and/or brightness are different for the WT and KO cells – this should be corrected. Second, I clearly see one WT cell with a much bigger Golgi, but for the other cells the difference to the KO cells is not obvious for me. In other words: I see also dispersed Golgi in some KO and compressed Golgi in some WT cells. Therefore, a proper quantification is crucial and the procedure, number of cells and selection of cells have to be described in detail.

Finally, I cannot judge the novelty in the VCP-cofactor complexes field. From a purely technical point there is no novelty.

Minor Points:

I believe that a change from 0.3 to 1.0 for the TR-FRET signal (Fig. 2) indicates a change in distance of around 5nm (depending on the Förster radius). How can this be explained by the deltaVCPID?

In the caption of Fig. 4b "First column" should read "First line" and so on.

Reviewer #3

(Remarks to the Author)

This manuscript is a resubmission of a manuscript first submitted in December 2022 in which the authors tried to address the comments made. Unfortunately, I am of the opinion that not all issues have been addressed appropriately and that the manuscript has not significantly improved during revision.

What this manuscript does well is describe the interaction of VCP with VCPIP1 through 2 interfaces: VCP-N with UBX-VCPIP1 and VCP-D2 with VCPIP1. The authors also show that cofactor p47 competitively binds via its UBX domain to VCP-N, competing with VCPIP1-UBX.

The main concern with the previous manuscript was the functional importance of these interactions. These are addressed by 2 different assays: one measuring deubiquitinating activity and the other Golgi reassembly.

- The authors assessed deubiquitinating activity using a Ub-Rho fluorescence assay, where ubiquitin cleavage results in rhodamine dequenching and increased fluorescence. While the results show slight differences among constructs and with/without VCP, the relevance of this assay to VCP-VCPIP1's actual function is questionable, Rho is not expected to interact with VCP. This would differ dramatically from the proposed role (as stated in the conclusion) of the VCPIP1-VCP complex in membrane fusion and especially in ensuring efficient Syn5 deubiquitylation where interaction between Syn5 and VCP is expected to take place. Nonetheless, an explanation of the impact of the interaction between VCPIP1 and VCP in the Rho-Ub assay would be beneficial.

-The authors assessed the importance of VCPIP1 domains in Golgi reassembly using fluorescence microscopy. Since the last manuscript, the authors changed the marker used to detect the Golgi, now using GM130. Using this more specific antibody to detect Golgi, a different phenomenon was "supposedly" observed: a difference in Golgi matrix size rather than 'gaps' in the Golgi. The conclusion remains the same, that "both VCPIP1's catalytic activity and its interaction with VCP were required for Golgi reassembly". The results of this experiment are still not convincing, and this for several reasons. Firstly, the authors rely on a 'rough measure of Golgi area' to conclude on Golgi reassembly. I would argue that Golgi surface area is not a direct measure of Golgi reassembly. To properly examine reassembly, it would be better to look at events that occur after disassembly, such as Golgi disassembly by incubation with Brefeldin A (BFA) and/or Golgi reassembly after mitosis (using synchronised cells and live imaging). Secondly, experiments are conducted using transient transfection, and the experimental workflow does not check that cells visualized have been transfected. Moreover, no quantification has been provided comparing Golgi size between cells transfected with WT versus domain mutants and catalytically inactive mutant. In any case, conclusions could not be reached without considering the transfection levels between wild-type (WT) protein and domain/deletion mutants.

Other issues remain with the manuscripts:

- Interpretation of stoichiometry of the VCP-VCPIP1 complex: While I agree that the data support that up to 3 VCPIP1 molecules can bind to VCP at the same time, it does not indicate the possibility and functionality of less binding at a given time. Moreover, the fact that imposing C6 symmetry improves the resolution of the UBX domain suggests that more VCPIP1 molecules may bind to a VCP complex at a given time: 3 via the VCPID motif and up to 6 via the UBX domain, and this is assuming that the 3 molecules interacting with the VCPID motif also interact with their UBX domains.

- While revisions have been made to the introduction of the manuscript, improvements are still required to remove frustration for the reader. Particular issues noted are confusion of results published with Cdc48 in yeast and p97 in humans, and inappropriate references being cited. It is important that authors check that the publications cited actually demonstrate the facts that they cite. Examples include:

- lines 34-38: " Structural and biochemical... how VCP unfolds ubiquitylated substrates...":

- citing Ref #20, p97:PP1, where unfolding does not require ubiquitination.

- many important studies have been omitted on 'binding partners and substrates', including work done with p47 (Shp1) and Ufd1-Npl4 in humans. While I agree that citing everything would be difficult, using a review would be more appropriate than citing Cdc48 work and examples of processes that are ubiquitination-independent.

- the authors should rephrase this section to include more concepts, including inhibitors, without any explanation.

- lines 41-42: "these cofactors have a VCP interacting motif such as UBX or PUB domain that interacts with substrates, in most cases 19,25". References 19 and 25 are misplaced as these studies do not demonstrate these interactions.

- Please note that these are examples, and the authors should check the accuracy of citations throughout the manuscript. Another example can be found in the discussion, line 352, where references 26 and 52 are inappropriately used as this review and study do not show that CB-5083 does not disrupt Golgi reassembly.

Others minor corrections required:

- Line 135: "allowed us to visualize the flexible nature of the OTU domain". This sentence needs to be rephrased because "flexibility" cannot be visualized.

- Line 161: "define residual binding"

- Line 175- 177: “)” “we note that the affinities of these experiments are about a factor of 3 different to the previously determined affinities”. Sentence needs to be rephrased.

REVIEWER COMMENTS

Reviewer #1 (Remarks to the Author):

Cellular functions of the hexameric ATPase VCP/p97/Cdc48 are controlled by a variety of regulatory cofactor proteins. Shah et al. solved the three-dimensional structure of VCP in complex with the deubiquitinase (DUB) cofactor VCPIP1 (also known as VCIP135) by cryo-EM, both in the absence and presence of an additional cofactor, p47. Previous work has shown that all three proteins cooperate in the post-mitotic reassembly of Golgi stacks. The VCP-VCPIP1 complex has a 6:3 stoichiometry, with one VCPIP1 protomer binding to two adjacent VCP protomers. Binding is mediated via two major interfaces: one is formed between VCPIP1's UBX domain and the VCP N-domain, and the other between the newly assigned, central VCPID domain of VCPIP1 and the bottom of VCP's D2 domain. Densities for VCPIP1's catalytic OTU domain and a stalk domain linking it to the VCPID domain are visible but were of insufficient quality for model building. Next, the authors determined the contributions of the UBX and VCPID domains to VCP binding and DUB activity, respectively, in biochemical assays. They found that the UBX domain was required for VCP binding, but not DUB activity, and vice versa for the VCPID domain. They also found that the DUB activity was stimulated by VCP binding. The structure of the ternary VCP-VCPIP1-p47 complex is very similar to the VCP-VCPIP1 complex, except that additional densities at VCP's N-domains indicated the binding of three UBX domains from p47 instead of two UBX domains from VCPIP1, suggesting competition of the two cofactors for binding to the N-domains. A binding competition experiment showed that p47 binds with about 3.5-fold higher affinity than VCPIP1 to VCP, which was attributed to the additional interaction between p47's SHP box and VCP's N-domain by the authors. To test the relevance of the VCPID and UBX domains for a known cellular function of VCPIP1, the authors complemented a VCPIP1 knockout cell line with plasmids encoding wild-type or mutant VCPIP1 and stained the Golgi apparatus. They concluded that the VCPID and UBX domains are both required for efficient Golgi reassembly.

The structural analysis of VCP-cofactor complexes is of high interest both from a basic research and biomedical perspective. Unfortunately, the present study of VCP-VCPIP1 complexes is compromised by its limited novelty compared to two recent papers by Liao et al (ref. 41, published) and Vostal et al (ref 40, preprint server). The overall structure of the VCP-VCPIP1 complex, the new binding interface between VCPID and the D2 domain of VCP, its importance for Golgi reassembly in cellulo, and the VCP-stimulated DUB activity of VCPIP1 were all reported in one or both of the competing studies. Vostal et al additionally solved the OTU domains of the VCP-VCPIP1 complex and proposed a model for the de-ubiquitination of substrate proteins at the VCP exit site. The structure of the ternary VCP-VCPIP1-p47 complex, reported for the first time in the present manuscript, provides only incremental new insight compared to the previous binary VCP-VCPIP1 and VCP-p47 structures. Moreover, parts of the structure-function analysis are incomplete, and the imaging data related to Golgi fragmentation are unconvincing.

We thank the reviewer for taking the time to review our manuscript and for their thoughtful insights, which we have addressed in the point-by-point response.

Specific points:

1. Fig. 2: There is clearly residual VCP binding for both VCPIP1 variants lacking the UBX domain, likely derived from interactions of the VCPID domain (cf. Liao et al, Vostal et al) and/or the SHP box recently identified by Nakayama & Kondo (res. 1018-1036; see ref. 32) - not shown in Fig. 1a. The authors should either quantify the residual binding using higher protein concentrations, or re-phrase their conclusions. Moreover, including the DUB activity of the isolated OTU domain could support the interpretation of the presented DUB data.

We agree with the reviewer that there may be residual binding. VCPIP1 SHP is now included in the VCPIP1 domain map. DUB activity of different constructs, including a construct lacking the VCPID and UBX domains, is included in EDF 3. Additionally, we have added “However, these constructs did have residual binding possibly from interactions with VCPIP1 VCPID and/or SHP box.” to the results section.

2. Fig. 3, Results lines 215ff.: The discussion of the competitive/simultaneous binding of VCPIP1 and p47 to VCP is incomplete. First, can the authors exclude the possibility that the three p47 UBX domains occupy N-domains spared by the VCPIP1 UBX domains?

Due to the dynamic nature of the VCP-VCPIP1-p47 system and considering the structural homology of the UBX domains, it is difficult to definitively isolate VCPIP1 UBX domain from p47 UBX domain bound to each VCP N-domain. We have tried to use the extra alpha helix characteristic of the p47 UBX domain to mask each N-domain with bound UBX individually in an effort to draw conclusions on how many VCPIP1 or p47 UBX domains are bound in complex, but unfortunately the occupancy of each protein is not clear due to the dynamic nature of the complex and precludes us from drawing any conclusions.

As the reviewer correctly points out, it is possible if not in fact very likely that simultaneous binding of VCPIP1 and p47 involves UBX domains from both binding to VCP N-domains. But given the inability to unambiguously demonstrate this with structural data we prefer to stay cautious on conclusions.

Second, if deletion of its UBX domain strongly decreases VCPIP1's affinity for VCP (Fig. 1), how can the ternary complex be stable in the absence of its UBX domain binding to VCP's N-domain?

We believe that VCPIP1 UBX interacts with the N-domain of VCP and apologize for not phrasing this point clearly and have attempted to clarify in the revised manuscript.

Our ternary complex includes both VCPIP1 and p47 (see **Extended Data Fig. 4a-b**). From our competition TR-FRET results (**Extended Data Fig. 4i**) we observed that VCPIP1 and p47 compete for binding, with p47 being a somewhat more effective binder (**Extended Data Fig. 4i**). Additionally, we can unambiguously assign density located at the VCP D2 region as VCPIP1 VCPID (**Fig. 3**). Moreover, although the density observed bound to the N domain of VCP is a better fit for p47 UBX, we suspect that VCPIP1 UBX binding is present in particles even though we are unable to clearly visualize it, likely because VCPIP1 and p47 are in dynamic equilibrium with respect to binding to VCP and share high similarity. The dynamic nature of VCP interactions with its adaptors is well-established, including the fact that many (if not all) adaptors bind to the same/similar region on VCP¹⁻⁴. There is evidence that some engage in a mutually non-exclusive manner, and we believe that this is the case here.

Third, the authors propose that the competition by/ higher affinity of p47 can be explained by additional contacts via its SHP box. Please note that VCPIP1 also has a SHP box (see above) that was shown to bind VCP (refs. 32, 41), and that p47 has a second SHP box (ref. 39), complicating the interpretation of competitive/simultaneous binding. More detailed insights would require the comparative analysis of mutations in all three SHP boxes.

We removed the SHP box analysis from the discussion. Further, the second SHP box and the SIM domain are also included in the p47 domain map.

3. Fig. 4: The Golgi staining in the different cells is not convincing at all. Much clearer images of Golgi fragmentation can be obtained by immunofluorescence staining using a GM130 antibody (see e.g. refs. 32, 41). The VCPIP1 WT and KO cells should be transfected with empty vector for a proper control. Quantification of Golgi fragmentation in at least three biological replicates is mandatory.

We thank the reviewer for the suggestions and have based on those entirely re-designed the cellular assay. The new cellular data uses the GM130 antibody with new empty vector controls in both A2058 WT and KO cells. We have optimized imaging conditions to obtain high quality confocal data and enable quantification. The data was collected with three biological replicates each with technical duplicates. Finally, quantification was done with approximately 55 cells in each replicate per condition and statistical analysis was done in GraphPad Prism with one-way ANOVA test ($p < 0.05$). The exact number of Golgi quantified for each condition has been added to the methods section.

4. Introduction, lines 39/40: Hemmo Meyer's work on ubiquitin-independent PP1 maturation by VCP-p37 should be mentioned.

We have added in a citation for Hemmo Meyer's work.

5. Results, line 218: The role of the SHP box in VCP binding was first reported in refs. 24 and 33.

We have added new citations.

6. Discussion, lines 291ff.: Please explain in more detail how the new structure "helps to rationalize effects of CB-5083". Please note that clinical trials of CB-5083 and CB-5339 had to be discontinued.

We thank the reviewer for pointing out the discontinuation of CB-5083 and CB-5339, which we now explicitly state in the manuscript. We have also added more details to how the structure helps to rationalize pharmacological observations, while also making it clear that this is just an example and how people use the new structural information is to be seen.

Reviewer #2 (Remarks to the Author):

In their study Shah et al. investigate ternary complex formation of the three proteins VCP, VCPIP1 and p47. The main conclusions are drawn from cryo-EM data. I will focus here on the time-resolved FRET (TR-FRET) and fluorescence data. The authors use trFRET to determine the K_D 's of several truncation mutants and to conclude that the UBX domain anchors VCPIP1 to VCP (Fig. 2). In principle I support their conclusion, but there remain a few questions on the experiments (see below). In addition, the authors conclude from the images in Fig. 4 that VCPIP1 and VCP interaction is necessary for Golgi reassembly. Here I cannot fully follow the argumentation. These are my specific questions:

1) Nomenclature: Why do the authors call the FRET-based method 'time-resolved'? I do not see any time-resolved data. If there is some, please show that.

We used the widely accepted name for this method to avoid confusion, which was first described in the late 1980s⁵. In brief, the experiment is called "time-resolved" although it is not actually based on real time-resolved measurements. The name "time-resolved" was given to this method because it uses a millisecond (50 to 70 ms) time-delay between excitation pulse and measure emission that allows short-lived background fluorescence to be "resolved" (*i.e.* eliminated)⁶⁻⁸.

2) Please define what the TR-FRET ratio is (*i.e.* the y-axis in Fig. 2). I even could not find this information in the main text of reference 60.

The TR-FRET ratio is calculated as the 520nm/490nm signal which are the detected emission signals for BODIPY and terbium, respectively. We have added this to the methods section. We have also renamed the axis to "520nm/490nm" to be exact.

3) Why does the TR-FRET ratio reach only about 0.3 for VCPIP1(WT) and almost 1.0 for VCPIP1(Δ VCPIP), although both have about the same affinity?

We thank the reviewer for asking this question. TR-FRET signal (or any FRET signal) is proportional to the sixth power of proximity between the donor and acceptor fluorophore and therefore extremely sensitive to conformational differences. Here, BODIPY-labeled VCPIP1 WT and VCPIP1 Δ VCPIP may be in different proximity to Tb-VCP resulting in different TR-FRET ratios.

4) Do the authors have any idea, why the DOL is so different (low) for VCPIP1(WT) compared to *e.g.* VCPIP1(Δ VCPIP)? I think that the BODIPY can react with any lysine and therefore should bind almost in a stoichiometric way?

The VCPIP1 WT and VCPIP1 Δ VCPIP protein samples are slightly different with varying numbers of free lysines on the cell surface, which could lead to variations in BODIPY-labeling. Generally, the DOL is only partially controllable with the non-saturating conditions used to avoid confounding effects from labeling all lysines and therefore we always determine it experimentally.

5) Fig. 4: Why were Z-stack max intensity projections used?

As explained in responses to reviewer #1 and 3 and also below, we have re-designed the cellular experiments and no longer use Z-stack max intensity projections.

6) Fig. 4: The white arrows are supposed to indicate “gaps” in the cytoplasm, the red ones “maximal intensity spots”. For both, I cannot see a difference between VCPIP1(WT) and the other cells in the cytoplasm. Could you please quantify what a “gap” and what a “maximal intensity spot” is and then give an estimate on how much a “more pronounced Golgi fragmentation” is?

We have redesigned the experiment with a revised methodology (see revised **Figure 4** and response to reviewer #1). With GM130 antibody staining, we are able to stain for the Golgi alone eliminating any background signal from the ER. Here, the images depict a dispersed Golgi phenotype in WT cells and a compressed Golgi phenotype in VCPIP1 KO cells. We analyzed three biological replicates with technical duplicates for each and were able to quantify the dispersion of Golgi seen in the images. Since GM130 stains for the protein that stabilizes the Golgi matrix, we believe that the dispersed phenotype demonstrates a fully reassembled Golgi while the compressed phenotype demonstrates the inability to reassemble Golgi.

7) What should I learn from the merged images in Fig. 4?

Merged images in **Fig. 4** allow the reader to see both DAPI and GM130 stain, which provides a visual representation of the Golgi stain in relation to the nucleus. It also ensures that there is one Golgi to one nucleus with no overlap. We hope this is clearer with the new images we collected.

Reviewer #3 (Remarks to the Author):

The manuscript reports work focused on the characterisation of VCP complexes involved in Golgi reassembly (VCP-VCPIP1 and VCP-VCPIP1-p47) using single particle cryo-EM and evaluating the functionality of the VCP-VCPIP1 interaction in cells.

Strengths of the manuscript include the detailed characterization of the mode of interaction between VCPIP1 and the D2 and N domains. This work represents a tour de force in image analysis, successfully extracting high-resolution information from highly heterogeneous complexes, likely affected by both conformational and compositional variability. The study also demonstrates that the UBX domain of VCPIP1 is competitively displaced by the UBX domain of p47. It further defines the detailed interaction between the p47 UBX-SHP linker and the N-domain of VCP. The cellular studies suggest that all domains of VCPIP1 play a critical role in Golgi reassembly.

Despite providing valuable insights, the manuscript comes across as rushed, with certain sections appearing incomplete and some conclusions lacking accuracy. The reason for this urgency becomes evident in the discussion section, where the authors mention two other manuscripts addressing similar work on the VCP-VCPIP1 complex. One is still in preparation (submitted to Bioarchives), while the other was published in *Advanced Science* in September 2024, shortly before this manuscript was submitted. While the pressure of such circumstances is understandable, it does not excuse the incomplete analysis of the work conducted or the failure to fully leverage its

potential. The manuscript faces challenges on two fronts: stylistic weaknesses and design/quality issues. Below, I will outline these shortcomings in detail.

Starting with the introduction, the authors describe the role of VCP in the ubiquitin-proteasome system (UPS), which feels unnecessary given the broader relevance of the topic to VCP's role in Golgi assembly. Moreover, the description is heavily biased, disproportionately emphasizing work published by Harvard groups while neglecting significant contributions from other researchers. It also conflates findings on VCP with those of its yeast homologs, leading to confusion and a lack of clarity.

We appreciate this comment and have extensively revised our introduction to address those points.

Regarding the analysis of the VCPIP1-VCP complex, the authors provide a detailed and well-described characterization of the interaction between the VCPIP1-UBX domain and VCP, as well as another domain referred to as VCPIP1-VCPID interacting with VCP. However, greater care should be taken when drawing conclusions about the 1:2 stoichiometry (VCPIP1:VCP), particularly given the extensive data excluded during the classification analysis, as shown in Extended Data Fig. 2. The conclusions are based on a subset of 233,953 particles, corresponding to only 38,992 particles considering the symmetry extension for the UBX domain and just 17,976 particles for VCPID- amounting to less than 3% of the particles analyzed.

We thank the reviewer for their insight. As seen in **Extended Data Fig. 2**, without C6 symmetry, a structure of VCP bound to three VCPIDs was solved, confirming the 1:2 stoichiometric relationship between VCPID and VCP. For these VCP-centric structures, C6 symmetry allowed us to account for VCPIP1 density that might be present at one of the six VCP N-domain positions. Some classes were not included for the final reconstruction since classes with the highest density were selected each round of classification. Considering that there is flexibility of the N-domain, there are other possible states of VCPIP1 UBX which were not selected for the final reconstruction, which had the N-domain in the down position. Therefore, there were other particles that included the VCP-VCPIP1 interaction but were excluded to account for conformational heterogeneity and to achieve the highest resolution in a given state. We have now included additional discussion about our analysis and its limitations.

Beyond the structural data, the nature of the VCPID interaction with two protomers of VCP supports the 1:2 stoichiometry on mechanistic grounds and we feel that the overall body of data supports the conclusion that the biologically relevant complex exists in a 1:2 stoichiometry.

The data appears to contain significantly more information than the authors choose to present or analyze. Notably, some of the classes clearly show densities above the D1 rings, potentially reflecting the upward motion of the N domain. However, this valuable information was lost in the consensus reconstruction, where all N domains are depicted as coplanar with the D1 ring. Similarly, this information is absent in the final D1-UBX density derived from less than 3% of the data.

Although the N-domain is seen to move up and down, the use of C6 symmetry expansion allowed us to optimize the density of VCPIP1 and to achieve the highest resolution of UBX density bound

to N-domain in the down position. To improve clarity with regards to data processing, and in some cases associated limitations, we have added a new section in the discussion.

While we agree that there is additional information in the data and spend significant time to extract as much as possible, we realized that the dynamic nature of the complex and the resulting poor reconstructions limits our ability to visualize additional states. We have opted to be conservative in our conclusions and what we present in the paper.

The resolved density of VCPIP1 interacting with the D2 domain is minimal in this study compared to the other two manuscripts mentioned above on VCPIP1. The provided movie is unconvincing in demonstrating the movement of these regions of VCPIP1; instead, it appears to reflect particle occupancy rather than dynamic motion. Have the authors considered performing focused classification specifically on VCPIP1, excluding VCP densities?

The movie demonstrating VCPID, its connecting stalk and corresponding OTU domain is from a 3D Variability job in cluster mode to prevent particle overlap between clusters. The 3D Variability Display job uses the data to compile each cluster which is used to create a movie in ChimeraX demonstrating dynamic motion. As evident from the movie, VCPID and the stalk region are highly flexible. Although the particles have the OTU domains, the number of particles with similar conformations is not enough to obtain a high-resolution structure not resulting in noisy density. Liao et al. has fit in an AlphaFold model in the noisy density of the OTU region they have deposited that lacks secondary structure resolution. While our map has similar noisy density, we don't feel comfortable fitting a model to it. In contrast, we have refined a structure with high enough resolution to de novo build our VCPIP1 models.

Ideally, we would want to perform focused refinement on specifically VCPIP1 without VCP. However, the resulting densities are not convincing, most probably since the Local Refinement job needs a larger mass to align on. Additionally, there might be flexibility in VCPIP1 itself. We found highest resolution density resulted from masking the D2 dimer with VCPID or VCP N and D1-domains with VCPIP1 UBX domain.

The subtitle referring to the VCP-VCPIP1-p47 complex is inaccurate, considering how little of the VCPIP1 and p47 proteins are resolved in the study. Similarly, the title "VCPIP1 catalytic activity and binding to VCP are required for Golgi reassembly" is also misleading, as the analysis primarily tested the effects of mutants lacking specific domains of VCPIP1, rather than mutants specifically targeting the interaction surfaces defined by the single-particle cryo-EM work.

While we agree with the reviewer that only the interacting domains are visible within the ternary structure, we respectfully disagree that the title is inaccurate. The complex we imaged is the complete VCPIP1-VCP-p47 complex reconstituted from full length recombinant proteins (see **Extended Data Fig. 4b**). We are open for suggestions on a better sub-title but as it stands feel this is the most accurate.

Regarding the second noted subtitle, we agree and have changed the title for the cellular assay to read "VCPIP1 catalytic activity and VCP interacting domains are required for Golgi reassembly" based off the recommendation.

The Golgi reassembly data presented using the full-length and domain deletion mutants is of overall poor quality, primarily due to its low resolution. This makes it difficult to distinguish between Golgi fragmentation (observed as speckles in the images, which are also present in VCPIP1 knockout cells with VCPIP1 reintroduced) and abnormal nuclear morphology. No quantification of the data has been attempted, and it is difficult to believe that the authors did not have access to higher-resolution light microscopes to more clearly demonstrate defects in Golgi stack reassembly. It is surprising that the authors did not utilize higher-resolution light microscopes to more clearly demonstrate defects in Golgi stack reassembly, nor did they attempt any quantification of the data.

We agree with the reviewer and this point has also been raised by reviewers #1 and 2. The limitation was not the microscope hardware, but rather the staining protocol used. Based on suggestions by reviewer #1, we have now re-designed the cellular assay using the GM130 antibody to remove background signal from the ER. This has allowed us to obtain higher resolution images of Golgi to accurately depict changes in phenotype with different VCPIP1 truncation mutations. Further, the improved staining protocol has now allowed us to quantify changes, and we have quantified biological triplicates each with technical duplicates. See also responses to reviewer #1 and 2.

Finally, and perhaps most importantly, no comparison has been made between the data presented in this manuscript and the findings of the manuscript published by Lao and colleagues in *Advanced Science* (reference 41). The only comment provided is that the other manuscript describes “some of the features... and further supports our conclusions,” which is insufficient for a meaningful analysis or discussion of overlapping results. There are also clear differences between these two studies, particularly in the binding of VCPIP1-UBX to VCP-N and the conformation adopted by the VCP N domain upon binding. These discrepancies warrant further discussion and analysis, as they could provide valuable insights into the mechanistic differences observed. Especially noteworthy is the fact that the two studies used different expression systems: mammalian cells in this study and bacteria in the *Advanced Science* manuscript. Interestingly, the nucleotide occupancy of VCP has been shown to differ depending on the expression system used. Specifically, VCP is nucleotide-free when expressed in mammalian cells, whereas it predominantly binds ADP when expressed in bacteria. This difference could significantly influence the observed interactions and conformations, further emphasizing the need for a detailed comparison between the two studies. The authors already have access to the necessary proteins and a beautiful TR-FRET assay, which would enable them to address these questions effectively and provide more comprehensive insights.

We thank the reviewer for these thoughtful comments and agree that while the studies are largely in agreement, there are some differences in observed conformations and other details. Some of these are due to our conservative approach to interpreting electron density and refraining from trying to fit atomic models into density that does not support model building. We would therefore like to refrain from contrasting those findings especially since there are also significant differences in approach to processing the data. The one area we found intriguing was the nucleotide state and thank the reviewer for reminding us of this. We have therefore now assessed VCPIP1 and VCP

interaction in the presence of different nucleotides (**Extended Data Fig, 6**). Additionally, we have added a brief discussion of how our study compares to the study of the other two groups.

References

1. Meyer, H., Bug, M. & Bremer, S. Emerging functions of the VCP/p97 AAA-ATPase in the ubiquitin system. *Nat Cell Biol* **14**, 117–123 (2012).
2. Yeung, H. O. *et al.* Insights into adaptor binding to the AAA protein p97. *Biochem Soc Trans* **36**, 62–67 (2008).
3. Schuberth, C. & Buchberger, A. UBX domain proteins: major regulators of the AAA ATPase Cdc48/p97. *Cell Mol Life Sci* **65**, 2360–2371 (2008).
4. Hänzelmann, P. & Schindelin, H. Characterization of an Additional Binding Surface on the p97 N-Terminal Domain Involved in Bipartite Cofactor Interactions. *Structure* **24**, 140–147 (2016).
5. Morrison, L. E. Time-resolved detection of energy transfer: Theory and application to immunoassays. *Analytical Biochemistry* **174**, 101–120 (1988).
6. Yue, H. *et al.* Diagnostic TR-FRET assays for detection of antibodies in patient samples. *Cell Rep Methods* **3**, 100421 (2023).
7. Ergin, E., Dogan, A., Parmaksiz, M., Elçin, A. E. & Elçin, Y. M. Time-Resolved Fluorescence Resonance Energy Transfer [TR-FRET] Assays for Biochemical Processes. *Curr Pharm Biotechnol* **17**, 1222–1230 (2016).
8. TR-FRET Measurements | BMG LABTECH. <https://www.bmglabtech.com/en/tr-fret/>.

REVIEWER COMMENTS

Reviewer #1 (Remarks to the Author):

The authors have satisfactorily addressed my concerns. The revised manuscript is suitable for publication in NCOMMS.

We thank the reviewer for all the constructive feedback.

Reviewer #2 (Remarks to the Author):

The authors have addressed my comments. In particular, they have now explained TR-FRET and re-designed the cellular fluorescence experiments.

We appreciate the feedback and are happy that we were able to address those points.

Nevertheless, there remains some ambiguity concerning the quantification of the area of Golgi (Fig. 4). I could not find a clear description on the quantification.

We thank the reviewer for this comment and apologize that our methods lacked sufficient detail. We have now expanded on our description of quantification in the methods section (see **line 803** and also here).

Quantification

A total of three biological replicates with two technical replicates each were imaged for the nine conditions, where each technical replicate was imaged on a separate day (six total days) with the same imaging settings. For visualization and quantification, images were imported into the imaging tool ImageJ's Fiji and brightness/contrast were chosen for a single image and set for the remaining images. Images were analyzed in reverse order in which they opened (last image was analyzed first). Next, a freeform selection was drawn around each Golgi that was fully visible in the field of view and the area of the freeform selection was calculated in Fiji by selecting the "Area" parameter in the *Set Measurement* tool and then using the *Measure* tool to calculate each Golgi selection. Areas of ~55 unique Golgi per replicate per condition were measured. Images in each replicate were analyzed and measurement concluded once ~55 Golgi per replicate per condition were quantified. The total number of Golgi quantified per condition is represented below.

Condition	Total # of Golgi quantified
WT	344
WT + EmptyVector	318
KO	356
KO + EmptyVector	341
KO + VCPIP1 WT	338
KO + VCPIP1 Δ VCPIP1	340
KO + VCPIP1 Δ UBX	364
KO + VCPIP1 Δ VCPIP1 Δ UBX	353
KO + VCPIP1 C219A	359

When I look at the images provided e.g. in Fig. 4c, I first notice that the contrast and/or brightness are different for the WT and KO cells – this should be corrected.

We believe that brightness/contrast should not be corrected to make images appear more similar. The apparent differences are the result of differences in the underlying samples, which are all imaged with the same imaging set-up and settings. We have now included this caveat and explanation in our figure legend. We further now provide a Google Drive link containing all raw images, so it is easier to relate the representative images to what the entire dataset looks like. GraphPad Prism raw data with all Golgi measurements is now also provided (see URL below).

Google Drive link:

<https://drive.google.com/drive/folders/1nL6UIvaJ32bQWimjOTnefVgbNUqSLTab?usp=sharing>

Second, I clearly see one WT cell with a much bigger Golgi, but for the other cells the difference to the KO cells is not obvious for me. In other words: I see also dispersed Golgi in some KO and compressed Golgi in some WT cells. Therefore, a proper quantification is crucial and the procedure, number of cells and selection of cells have to be described in detail.

We completely agree that a proper quantification is crucial, which is why we have conducted it meticulously (**see Figure 4**) and apologize that was not clear from our description. As mentioned above, we now also provide the raw dataset so it is clear that no conclusions were drawn from individual images.

To provide some more background, before implementing our current procedure, we tried automated quantification using CellProfiler. However, we encountered the issue of incorrect assignment of Golgi sections to the correct nucleus, even after manipulating *blur* and *focus* parameters in CellProfiler. Therefore, and consistent with other recent publications, we performed manual quantification of Golgi area as a proxy to Golgi health/re-assembly and to ensure robustness, we increased the number of samples, including proper controls, and replicates. We aimed to analyze about 55 cells per replicate per condition for a total of approximately 330 Golgi per condition (exact number of Golgi quantified per condition was mentioned in methods and pasted above, **see line 803**). To be able to quantify this number of cells per condition, we collected ~15 images with an observed variation of 2 - 7 Golgi per image per condition. The images were all opened in Fiji and a freeform mask was drawn around each Golgi that was fully visible in the image. Partially visible Golgi were excluded from the analysis, and quantification for each replicate concluded when the first ~55 cells were analyzed – the remaining Golgi were not accounted for in the quantification resulting in a few unused images due to variability of number of Golgi in each field of view. For data transparency, all data points representing Golgi area are represented in the bar graph demonstrating the range of areas observed for each condition. This biological variance is captured in the representative images (Fig. 4) for the data set, and we now also provide the full set of images/GraphPad Prism raw data file (see above).

Finally, I cannot judge the novelty in the VCP-cofactor complexes field. From a purely technical point there is no novelty.

VCP has at least ~30 adaptor proteins that are known to bind mostly to the N-terminal end with a handful of adaptors that bind to the C-terminal end. Due to the ubiquitous role VCP plays in the cell, many studies aim to understand how VCP is localized to different parts of the cell and how its specific function is determined within a biological process. Previous studies have demonstrated that these adaptor proteins are critical in localizing VCP and specifying its function. VCPIP1 and p47 are adaptor proteins that were thought to bind to VCP, but it remained unclear how they interacted with VCP, and if they were able to form a ternary complex. In this paper, we determine high resolution structures to characterize the interaction of VCPIP1 with VCP, as well as in complex with p47.

While the structure of the VCP-VCPIP1 complex has been reported in two other parallel papers^{1,2}, our work still provides details not provided by these contemporary studies. From a strictly structure perspective, through our biochemical reconstitutions and extensive data processing (see **Extended Data Fig. 2 and 5**), we deposit eight maps and six models in the PDB and EMDb databases that all provide novel information.

- Model of de novo built 3x VCPID bound to VCP
- Model of de novo built VCPIP1 UBX domain bound to VCP N-domain
- Model of p47 bound to VCP N-domain
- Map of ternary complex of VCP-VCPIP1-p47 complex
- Visual depiction of VCPIP1 movement at the C-terminal end of VCP

Our paper also for the first time presents structural data of the ternary complex containing both, VCPIP1 and p47 simultaneously bound to VCP. We evaluate the structural homology of p47 and VCPIP1 UBX domains and its effect on binding affinities to the N-domain of VCP. This is significant as it establishes that 1) multiple co-factors can utilize N-domains to simultaneously bind to VCP and 2) p47 and VCPIP1 form a physical complex with VCP establishing the VCP species relevant for Golgi reassembly.

Notably, we functionally characterize the interacting domain of VCPIP1 at the C-terminal end of VCP, VCPID, as a critical domain for DUB activity. Additionally, we place the VCP-VCPIP1 complex in context of Golgi reassembly by solving the structure of the complex with VCP adaptor protein, p47, and interrogating the importance of VCPIP1 domains, UBX domain and VCPID, and its DUB activity (C219) in assembly using cellular assays with a VCPIP1 KO cell line.

Minor Points:

I believe that a change from 0.3 to 1.0 for the TR-FRET signal (Fig. 2) indicates a change in distance of around 5nm (depending on the Förster radius). How can this be explained by the deltaVCPID?

The reviewer is correct in that a change in TR-FRET signal relates to a change in distance with an inverse 6th-power law. However, our assay was never intended or designed to

accurately measure distance or changes in conformations as we use non-site specific labeling with lysine reactive fluorophores. Therefore, the distribution of label can differ from mutant to mutant.

Instead, our assay was simply designed to measure equilibrium binding by fitting a curve against the dose response signal. The TR-FRET signal in this case becomes proportional to the fraction bound and the absolute values irrelevant.

Given this assay design, we prefer to not speculate about any underlying reasons for the difference in TR-FRET signal. We have now made this clarification in the methods section.

In the caption of Fig. 4b “First column” should read “First line” and so on.

The changes have been made in the caption and now reads:

Figure 4: VCPIP1 catalytic activity and interaction with VCP contribute to Golgi assembly. (a) Western blot, blotting for VCPIP1, demonstrating successful transient transfection of VCPIP1 constructs in A2058 VCPIP1 KO cells. (b) Representative fluorescent images of A2058 WT and KO cells in which various VCPIP1 constructs were transiently transfected into the cells. Cells were stained with DAPI (first row) and GM130 antibody (second row). Both channels were merged as depicted in the third row. Scale bar is 10 μm (depicted in white). Images were taken on different days using the same imaging set-up, which may result in differences in apparent brightness. (c) Representative freeform mask (yellow) drawn in ImageJ Fiji around Golgi signal from one cell from a single merged image from WT and KO conditions are shown. (d) Quantification of area of Golgi per cell from various conditions are represented in a GraphPad Prism column chart. The data represented as mean \pm SD from three biological replicates each with technical duplicates. The p-values were calculated using one-way ANOVA between KO and all other conditions using GraphPad Prism software ($p < 0.05$).

Reviewer #3 (Remarks to the Author):

This manuscript is a resubmission of a manuscript first submitted in December 2022 in which the authors tried to address the comments made. Unfortunately, I am of the opinion that not all issues have been addressed appropriately and that the manuscript has not significantly improved during revision.

We appreciate the reviewer’s comments and thoughts and below, we are now providing additional details to address the remaining points. We would like to point out that during our first revision, we included new biochemical data sampling various nucleotides while assessing binding affinities of VCPIP1 and VCP. Additionally, we provided new, robust cellular data with GM130 antibody stain and quantification data from analyzing three biological replicates with technical duplicates for each replicate. These are two major new experiments that we believe addressed the key concerns expressed by all three reviewers and significantly improved the revised manuscript. We also believe that we have addressed the remaining comments below.

What this manuscript does well is describe the interaction of VCP with VCPIP1 through 2 interfaces: VCP-N with UBX-VCPIP1 and VCP-D2 with VCPIP1. The authors also show that cofactor p47 competitively binds via its UBX domain to VCP-N, competing with VCPIP1-UBX.

We are pleased to hear that the reviewer appreciates our structural and mechanistic dissection of the complexes.

The main concern with the previous manuscript was the functional importance of these interactions. These are addressed by 2 different assays: one measuring deubiquitinating activity and the other Golgi reassembly.

- The authors assessed deubiquitinating activity using a Ub-Rho fluorescence assay, where ubiquitin cleavage results in rhodamine dequenching and increased fluorescence. While the results show slight differences among constructs and with/without VCP, the relevance of this assay to VCP-VCPIP1's actual function is questionable, Rho is not expected to interact with VCP. This would differ dramatically from the proposed role (as stated in the conclusion) of the VCPIP1-VCP complex in membrane fusion and especially in ensuring efficient Syn5 deubiquitylation where interaction between Syn5 and VCP is expected to take place. Nonetheless, an explanation of the impact of the interaction between VCPIP1 and VCP in the Rho-Ub assay would be beneficial.

We agree with the reviewer's concern of using ubiquitylated rhodamine as a biological substrate to determine VCP-VCPIP1 function. However, our goal for using the Ub-Rho assay was to assess DUB activity of VCPIP1 when bound to VCP. We believe that the use of an artificial substrate, such as Ub-Rho, is informative in such an experiment since it excludes contributions from VCP that may occur with a more physiological substrate. We have made changes to the manuscript to be more explicit about this intent (see lines 196-198) .

-The authors assessed the importance of VCPIP1 domains in Golgi reassembly using fluorescence microscopy. Since the last manuscript, the authors changed the marker used to detect the Golgi, now using GM130. Using this more specific antibody to detect Golgi, a different phenomenon was "supposedly" observed: a difference in Golgi matrix size rather than 'gaps' in the Golgi. The conclusion remains the same, that "both VCPIP1's catalytic activity and its interaction with VCP were required for Golgi reassembly". The results of this experiment are still not convincing, and this for several reasons.

Firstly, the authors rely on a 'rough measure of Golgi area' to conclude on Golgi reassembly. I would argue that Golgi surface area is not a direct measure of Golgi reassembly. To properly examine reassembly, it would be better to look at events that occur after disassembly, such as Golgi disassembly by incubation with Brefeldin A (BFA) and/or Golgi reassembly after mitosis (using synchronised cells and live imaging).

The reviewer brings up a valid concern in that while the change in surface area of Golgi is most likely due to the changes in Golgi assembly, other factors cannot be ruled out.

Our main goal with these experiments was to assess whether deleting specific domains of VCPIP1 would lead to Golgi phenotypes in line with what has been observed with VCPIP1 deletions in prior literature in the field. We did not design the experiments in a way that would allow us to mechanistically dissect how VCPIP1 loss affects Golgi exactly and we apologize if that has not been clearly communicated. While we hence agree that our assay does not capture the level of detail to mechanistically dissect the effects on Golgi, we do think that the quantification employed accurately reflects an impact on Golgi that is likely due to defects in re-assembly based on prior literature. Precisely understanding the mechanism by which defects in VCPIP1 lead to such Golgi defects is beyond the scope of this work. We have modified text in the manuscript to better reflect these limitations.

Secondly, experiments are conducted using transient transfection, and the experimental workflow does not check that cells visualized have been transfected.

Transient transfection efficiency was assessed by western blot (anti-VCPIP1) for one biological replicate (see **Fig. 4a** and methods). The next two replicates were transiently transfected using the same protocol.

Moreover, no quantification has been provided comparing Golgi size between cells transfected with WT versus domain mutants and catalytically inactive mutant. In any case, conclusions could not be reached without considering the transfection levels between wild-type (WT) protein and domain/deletion mutants.

Quantification comparing Golgi area was completed with biological triplicates and technical duplicates each as per Reviewer #3's suggestions from previous revisions (see **Fig. 4d**, line 298 in main text and line 803 in methods). Please note expanded description of quantification methods compared to previous review, as described in detail above.

Although transfection levels are not the same across all conditions in the western blot (**Fig 4a**), we observe successful transfection for each construct. More importantly, expression levels are higher for some mutants that show defects over wild type controls, which if anything should blunt any observed phenotype. We agree that conclusions would be difficult if expression of mutants that lack rescue would be reduced compared to wild type, but given the opposite directionality we feel comfortable in our results.

Other issues remain with the manuscripts:

- Interpretation of stoichiometry of the VCP-VCPIP1 complex: While I agree that the data support that up to 3 VCPIP1 molecules can bind to VCP at the same time, it does not indicate the possibility and functionality of less binding at a given time. Moreover, the fact that imposing C6 symmetry improves the resolution of the UBX domain suggests that more VCPIP1 molecules may bind to a VCP complex at a given time: 3 via the VCPID motif and up to 6 via the UBX domain, and this is assuming that the 3 molecules interacting with the VCPID motif also interact with their UBX domains.

We thank the reviewer for bringing up this important point about stoichiometry and agree that while our structure supports binding of up to 3 VCPID domains at the same time and possibly up to 6 VCPIP1 engaged via UBX domains, although we believe that a 2:1 overall stoichiometry in our biochemical reconstitutions is likely (see below), in the context of a dynamic system and competitive binding as with VCP the structure only represents a snapshot. We have rephrased multiple parts of the manuscript to be very explicit about this limitation and that future studies (enabled by our structural characterization) are necessary to determine the exact nature of the physiological VCP-VCPIP1-p47 complex (see lines 144-146, 319-325)

While not necessarily relevant to the question of physiological stoichiometries as outlined above, we would still like to briefly summarize why we do not agree that processing using symmetry expansion the way we did necessarily implies higher occupancy.

Revisions Figure 1: Imposing symmetry vs. symmetry expansion. Particle imposed with C6 symmetry (first row) with extra density indicated by black dot. Particle is rotated on the C6 axis resulting in averaging and 1/6th extra density (grey) at every rotation. Second row demonstrates an expansion of the particle set by 6x, rotated at the C6 axis.

In our processing, we are not “imposing” C6 symmetry expansion, rather we are symmetry expanding our data set 6x. Imposing C6 symmetry averages the particle density – for example, UBX domain (black) - across six positions (input = 1 particle, output = 1 particle, shown in grey) whereas C6 symmetry expansion copies and rotates the particle on the C6 axis (input = 1 particle, output = 6 particles) enabling us to compile all the data from all six positions into one position (**see Revisions Fig. 1**). Next, we use tools such as 3D variability and 3D classification, that do not use alignment, with a mask for focused refinement to examine if we can resolve more information about the density of interest (UBX domain or VCPID) in the exact location of the mask and remove negative signal to increase our signal-to-noise ratio (see colored vs. grey densities in **Extended Data Fig. 2**). The benefit of 3D classification in cryoSPARC is that the job runs quickly and can sort out differences in classes without any alignment – we intentionally avoid using homogenous refinement after a symmetry expansion job for this reason. It is important to note that if we had not used symmetry expansion, it is possible we would have seen density for the UBX domain in all six positions since alignment would be based on the largest portion of density which in this case is VCP, rather than the UBX domain of VCPIP1 (**see Revisions Fig. 2 below**). This would have also resulted in a lower particle count in one location leading to a lower resolution structure.

Examples of 6 particle orientations
without symmetry expansion

Revisions Figure 2: Particle orientation without symmetry expansion. Using a hypothetical example of density of two VCPIP1 UBX domains (black dot) that can be seen at every rotation. If these particles were stacked on top of each other with no alignment, extra density would be observed in every position.

- While revisions have been made to the introduction of the manuscript, improvements are still required to remove frustration for the reader. Particular issues noted are confusion of results published with Cdc48 in yeast and p97 in humans, and inappropriate references being cited. It is important that authors check that the publications cited actually demonstrate the facts that they cite. Examples include:

- lines 34-38: “ Structural and biochemical... how VCP unfolds ubiquitylated substrates...”: citing Ref #20, p97:PP1, where unfolding does not require ubiquitination.

We appreciate the reviewer’s insight regarding the citations. We have made the necessary changes as requested. Citation #20 was added as per prior request and since has been removed from this line.

- many important studies have been omitted on 'binding partners and substrates', including work done with p47 (Shp1) and Ufd1-Npl4 in humans. While I agree that citing everything would be difficult, using a review would be more appropriate than citing Cdc48 work and examples of processes that are ubiquitination-independent.

We have added more citations in this section including work on p47 and UN.

- the authors should rephrase this section to include more concepts, including inhibitors, without any explanation.

We apologize, but we are unable to identify the section the reviewer is referring to.

- lines 41-42: “these cofactors have a VCP interacting motif such as UBX or PUB domain that interacts with substrates, in most cases 19,25”. References 19 and 25 are misplaced as these studies do not demonstrate these interactions.

We have added additional citations that further demonstrate these interactions. It now reads “Traditionally, these cofactors have a VCP interacting motif such as UBX or PUB domain that interacts with substrates, in most cases³⁻⁸.”

- Please note that these are examples, and the authors should check the accuracy of citations throughout the manuscript. Another example can be found in the discussion, line 352, where references 26 and 52 are inappropriately used as this review and study do not show that CB-5083 does not disrupt Golgi reassembly.

We agree and have now changed the wording and reassigned citations in the manuscript. It now reads “CB-5083 preferentially targets and has high specificity for the D2 domain of VCP⁹⁻¹¹ (IC₅₀ = 11 nM)^{10,12}. Interestingly, it has been shown that the binding of p47 to VCP decreases the potency of a D2 domain targeted inhibitor ~50 fold¹³. In contrast, an inhibitor that targets both the D1 and D2 domain has a ~4-6 fold decrease in potency with p47 bound¹³ demonstrating inter domain communication¹¹, and possible downstream effects on cellular functions including Golgi assembly¹¹.”

Others minor corrections required:

- Line 135: “allowed us to visualize the flexible nature of the OTU domain”. This sentence needs to be rephrased because “flexibility” cannot be visualized.

Sentence was changed to “this allowed us to visualize the movement of the OTU domain”.

- Line 161: “define residual binding”

Residual binding refers to non-specific binding. We have clarified this in the manuscript.

- Line 175- 177: “)” “we note that the affinities of these experiments are about a factor of 3 different to the previously determined affinities”. Sentence needs to be rephrased.

Sentence was rephrased and now states, “we note that there are differences between the apparent affinities of these experiments to the previously determined affinities attributed to different preparations of the proteins leading to batch effects but not impacting within experiment comparisons.”

References

1. Liao, T. *et al.* Molecular Basis of VCPIP1 and P97/VCP Interaction Reveals Its Functions in Post-Mitotic Golgi Reassembly. *Advanced Science* **11**, 2403417 (2024).
2. Vostal, L. E., Dahan, N. E., Reynolds, M. J., Kronenberg, L. I. & Kapoor, T. M. Structural insights into the coupling between VCP, an essential unfoldase, and a deubiquitinase. *Journal of Cell Biology* **224**, e202410148 (2025).
3. Raman, M. *et al.* Systematic proteomics of the VCP–UBXD adaptor network identifies a role for UBXN10 in regulating ciliogenesis. *Nat Cell Biol* **17**, 1356–1369 (2015).
4. Xue, L. *et al.* Valosin-containing protein (VCP)–Adaptor Interactions are Exceptionally Dynamic and Subject to Differential Modulation by a VCP Inhibitor. *Mol Cell Proteomics* **15**, 2970–2986 (2016).
5. Alexandru, G. *et al.* UBXD7 binds multiple ubiquitin ligases and implicates p97 in HIF1alpha turnover. *Cell* **134**, 804–816 (2008).
6. Buchberger, A., Schindelin, H. & Hänzelmann, P. Control of p97 function by cofactor binding. *FEBS Letters* **589**, 2578–2589 (2015).
7. Schuberth, C. & Buchberger, A. UBX domain proteins: major regulators of the AAA ATPase Cdc48/p97. *Cell Mol Life Sci* **65**, 2360–2371 (2008).
8. Elsasser, S. & Finley, D. Delivery of ubiquitinated substrates to protein-unfolding machines. *Nat Cell Biol* **7**, 742–749 (2005).
9. Kilgas, S. & Ramadan, K. Inhibitors of the ATPase p97/VCP: From basic research to clinical applications. *Cell Chemical Biology* **30**, 3–21 (2023).
10. Cheng, C. *et al.* VCP/p97 inhibitor CB-5083 modulates muscle pathology in a mouse model of VCP inclusion body myopathy. *Journal of Translational Medicine* **20**, 21 (2022).

11. Chou, T.-F. *et al.* Specific Inhibition of p97/VCP ATPase and Kinetic Analysis Demonstrate Interaction between D1 and D2 ATPase domains. *Journal of molecular biology* **426**, 2886 (2014).
12. Zhou, H.-J. *et al.* Discovery of a First-in-Class, Potent, Selective, and Orally Bioavailable Inhibitor of the p97 AAA ATPase (CB-5083). *J. Med. Chem.* **58**, 9480–9497 (2015).
13. Fang, C.-J. *et al.* Evaluating p97 Inhibitor Analogues for Their Domain Selectivity and Potency against the p97–p47 Complex. *ChemMedChem* **10**, 52–56 (2015).